# Forecasting the outcome of spintronic experiments with Neural Ordinary Differential Equations

Xing Chen[1,2], Flavio Abreu Araujo [3,4], Mathieu Riou[4], Jacob Torrejon[4], Dafiné Ravelosona[2], Wang Kang[1], Weisheng Zhao [1], Julie Grollier [4] & Damien Querlioz [2✉]

Deep learning has an increasing impact to assist research, allowing, for example, the discovery of novel materials. Until now, however, these artificial intelligence techniques have fallen short of discovering the full differential equation of an experimental physical system. Here we show that a dynamical neural network, trained on a minimal amount of data, can predict the behavior of spintronic devices with high accuracy and an extremely efficient simulation time, compared to the micromagnetic simulations that are usually employed to model them. For this purpose, we re-frame the formalism of Neural Ordinary Differential Equations to the constraints of spintronics: few measured outputs, multiple inputs and internal parameters. We demonstrate with Neural Ordinary Differential Equations an acceleration factor over 200 compared to micromagnetic simulations for a complex problem – the simulation of a reservoir computer made of magnetic skyrmions (20 minutes compared to three days). In a second realization, we show that we can predict the noisy response of experimental spintronic nano-oscillators to varying inputs after training Neural Ordinary Differential Equations on five milliseconds of their measured response to a different set of inputs. Neural Ordinary Differential Equations can therefore constitute a disruptive tool for developing spintronic applications in complement to micromagnetic simulations, which are time-consuming and cannot fit experiments when noise or imperfections are present. Our approach can also be generalized to other electronic devices involving dynamics.

[1] Fert Beijing Institute, MIIT Key Laboratory of Spintronics, School of Integrated Circuit Science and Engineering, Beihang University, 100191 Beijing, China. [2] Université Paris-Saclay, CNRS, Centre de Nanosciences et de Nanotechnologies, Palaiseau, France. [3] Institute of Condensed Matter and Nanosciences, Université catholique de Louvain, Place Croix du Sud 1, Louvain-la-Neuve 1348, Belgium. [4] Unité Mixte de Physique, CNRS, Thales, Université Paris-Saclay, Palaiseau, France. ✉email: damien.querlioz@c2n.upsaclay.fr

By combining spin and charge degrees of freedom, spintronics offers multiple functionalities that are exploited in industrial applications for sensing and memory storage[1–4], and are currently being studied for communications[5] and information processing[6–11]. The rich functionality of spintronic devices stems from the intricate magnetic textures from which they are formed, and the complex dynamical modes that can be excited in these textures. Spintronic systems, which have typical dimensions of a few nanometers to a few micrometers, cannot indeed be considered as formed by a single spin and have a large number of hidden variables: all the local magnetizations in the device. These spin textures can be dynamically excited by a wealth of physical quantities: magnetic fields, electrical currents or voltages, temperature, and pressure, all of which giving rise to different responses.

The dominant approach to predicting the complex behavior of spintronic devices is micromagnetic simulations. They divide the structures into nanometer-sized cells and simulate the spin dynamics of each cell using the Landau-Lifshitz-Gilbert equation, taking into account local and non-local interactions between the micromagnetic cells[12–16]. This technique, therefore, involves a considerable number of coupled differential equations and requires very long simulation times, easily reaching weeks in time-dependent experiments or in micrometer-scale devices. Beyond their long simulation time, micromagnetic simulations come with essential limitations. The simulations have to be re-executed from scratch when the input parameters of the template need to be modified. Also, micromagnetic simulations can almost never fit quantitatively the results of an experiment. In a real experiment, the geometry of a nanostructure is indeed always approximate, the material parameters can never be perfectly controlled and may possess specific structural inhomogeneities. Experimental results are also easily affected by the injection of noise, the details of the measurement setups, and unknown external factors, which are challenging to consider in the micromagnetic modeling process. A new tool that could accurately predict experiments, even when all these non-idealities are present, would be invaluable. For example, experiments in the field of neuromorphic spintronics[7,17,18] currently involve months-long experimental campaigns to optimize all the inputs of the systems, a development time that could be reduced radically with an appropriate modeling tool. In industry, the development of spin-torque magnetoresistive memory (ST-MRAM) also involves a considerable amount of micromagnetic simulations and experiments to optimize device parameters[19].

The progress of artificial neural networks provides an alternative road to simulate the behavior of spintronic systems and predict the results of experiments. In recent years, machine learning has been increasingly used in physics, for example, for discovering new materials and learning physical dynamics from time-series data.[20–30]. In the field of nanomagnetism and micromagnetics, deep neural networks are used to extract microstructural features in magnetic thin film elements[31–34], and to explore materials with ease[35]. Refrences[36–38] use a sophisticated combination of machine learning techniques to predict the magnetization dynamics of magnetic thin film elements over one nanosecond. However, the power of artificial neural networks has never been used to model, fit and forecast the long-term experimental behavior of solid-state nanocomponents. In this context, a recent type of neural network, Neural Ordinary Differential Equations (ODE), has great potential for modeling physical nanodevices, as it is specialized in predicting the trajectories of dynamical systems (Fig. 1c).

Neural ODEs, initially introduced in ref. [39], are ODE models $\dot{\mathbf{y}} = f_\theta(\mathbf{y}, t)$, where the function $f$ is expressed by a neural network with parameters $\theta$, which, instead of being explicitly defined, can be learned in a supervised manner. The machine learning process identifies the $\theta$ values that allow the Neural ODE to reproduce presented trajectory examples (training dataset), through the stochastic gradient descent algorithm. Once the Neural ODE has been properly trained on the training data, the corresponding equation becomes an appropriate model of the system dynamics and can be used to predict its behavior in novel situations not included in the training dataset.

Unfortunately, in their original form, Neural ODEs cannot be applied to the simulation of spintronic systems and solid-state devices in general, due to two major challenges:

- Neural ODEs require measuring the evolution of all the system variables, whereas in experiments and most applications, a single physical quantity is typically measured.
- Neural ODEs are not designed for dealing with external time-varying inputs.

In this work, we solve both issues and show that Neural ODEs can accurately predict the behavior of a non-ideal nanodevice, including noise, after training on a minimal set of micromagnetic simulations or experimental data, with new inputs and material parameters, not belonging to the training data.

In the rest of the paper, we first explain how we modified Neural ODE in order to be able to train the whole set of parameters based on the temporal evolution of a single physical variable of the nanodevice under the effect of fluctuating inputs. For this purpose, we have integrated in the Neural ODE framework the idea of the embedding theorem for the reconstruction of the state space from a time series. We then compare in detail the results obtained by this method with micromagnetic simulations. We demonstrate that Neural ODEs can accurately predict the complex evolution of a skyrmion-based reservoir computer, in a significantly reduced time compared to micromagnetic simulations (20 min versus 3 days). Finally, we demonstrate that this state-of-the-art deep learning technique for time series modeling can be applied to complex real-world physical processes. We train Neural ODEs to predict the results of real experiments on spin-torque nano-oscillators. These experiments would be impossible to model with micromagnetic simulations, as they would require hundreds of years of simulation. Our results show that, on the other hand, Neural ODEs quickly and accurately predict the outcome of experiments, including the associated noise.

## Results

To introduce the use of Neural ODEs in spintronics, we consider a device made of a skyrmion, a chiral spin texture extensively studied today for its fascinating topological properties, as well as its stability, compact size, and non-volatility, all of high interest for applications[40–42] (Fig. 1a). We consider the device of Fig. 1b, with two inputs: the perpendicular magnetic anisotropy (PMA) constant $K_u$ and the Dzyaloshinskii-Moriya interaction (DMI) $D$. The output of the device is the average magnetization perpendicular to the thin film axis, $\Delta m_z$, which translates directly to the electrical resistance of the device. In experiments, the PMA may be modulated by voltage through voltage-controlled magnetic anisotropy (VCMA) effects, while the DMI is typically a constant of the material. However, to train a Neural ODE, we perform micromagnetic simulations where these two quantities vary artificially with random sine variations, during 50 ns, to explore the possible responses that the system can exhibit (Fig. 1d). Figure 1e shows the elaborate variations that the output $\Delta m_z$ follows in these conditions. Our goal is to use this 50-nanoseconds time trace, which can be obtained in 40 min of micromagnetic simulations, to train a Neural ODE, capable of predicting the behavior of the system in any new situation, and in particular on long times whose simulation would take days with micromagnetics.

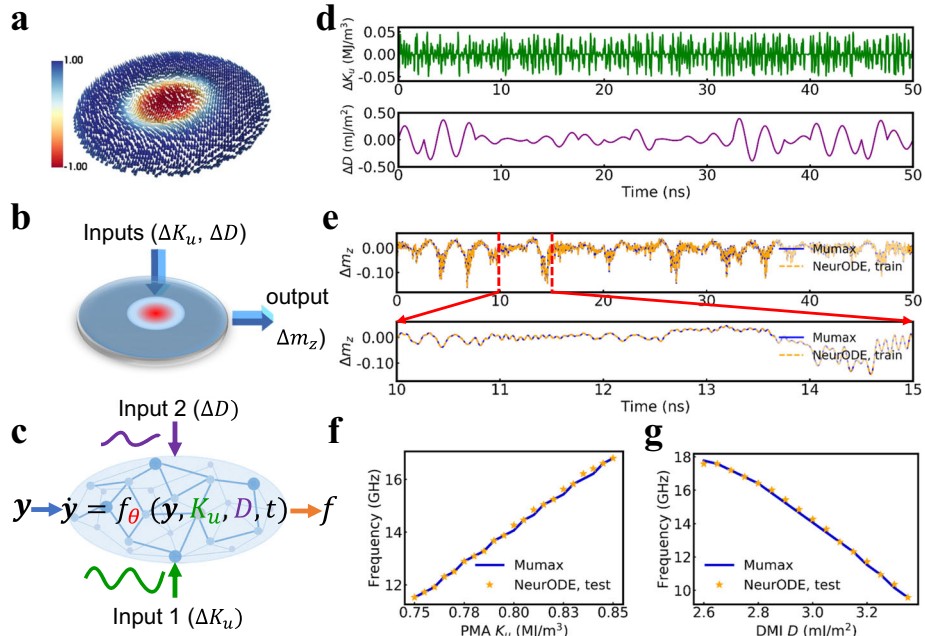

**Fig. 1 Modeling a skyrmion-based device with micromagnetic simulation and Neural ODEs. a** Magnetic skyrmion configurations in a nano-disk. The color scale represents the out-of-plane component magnetization, and arrows denote the spin orientation. **b** Sketch of a device where a single skyrmion exists in the ferromagnetic layer. The behavior of the device depends on the Perpendicular Magnetic Anisotropy (PMA) constant (e.g., through VCMA effect) $K_u$ and the Dzyaloshinskii-Moriya Interaction (DMI) strength $D$. The output signal is the variation of perpendicular component of the mean magnetization $\Delta m_z$. **c** Sketch of a Neural ODE structure $\dot{\mathbf{y}} = f_\theta(\mathbf{y}, K_u, D, t)$ with $\Delta K_u$ and $\Delta D$ as external inputs into the neural network. $\mathbf{y}$ is a vector of system dynamics and $f$ is defined by a neural network (see Fig. 2 for details of the modeling method). **d** Time-dependent random sine variation $\Delta K_u$ ranging from $-0.05$ MJ/m$^3$ to $0.05$ MJ/m$^3$ as an input (Input 1) and random sine variation $\Delta D$ ranging from $-0.4$ mJ/m$^2$ to $0.4$ mJ/m$^2$ as another input (Input 2) applied to the skyrmion system. Equilibrium values of $K_u = 0.8$ MJ/m$^3$ and $D = 3$ mJ/m$^2$ are used. **e** Predicted training output of $\Delta m_z$ by a Neural ODE in comparison with micromagnetic simulation (Mumax) results as a function of time. **f, g** Test results of the trained Neural ODE. The intrinsic response frequency of the skyrmion system for different values of $K_u$ in **f** and of $D$ in **g** calculated by using the trained Neural ODE (orange star) and by micromagnetic simulations (blue). 'Neural ODE' is abbreviated to 'NeurODE' in the legends of the figures.

**Extension of the Neural ODEs formalism to deal with incomplete information of dynamics.** Neural ODEs take the conventional form of ordinary differential equations $\dot{\mathbf{y}} = f_\theta(\mathbf{y}, t)$, but where the function $f_\theta$ is a neural network (Fig. 1d). The vector $\theta$ contains the parameters of this neural network, i.e., its synaptic weights and neuron thresholds. The vector $\mathbf{y}(t)$ describes the different state variables of the system: the function $f_\theta$ is therefore a neural network that takes $\mathbf{y}$ as input and provides the derivative $\dot{\mathbf{y}}$ as output. Once an initial value of $\mathbf{y}$ is given, the system dynamics is computed automatically by calling an ODE solver. Training a Neural ODE model, i.e., optimizing the $\theta$ parameters, normally requires the knowledge of the evolution of all these state variables over a collection of demonstrative examples[39]. After the training process has been completed, the Neural ODEs can be used to predict unseen data.

This conventional technique for training Neural ODEs has strong limitations for predicting the behavior of physical systems. It is often impossible to know all the state variables relevant to the dynamics of a physical system. For example, in the spintronic structure of Fig. 1b, only the mean magnetization $\Delta m_z$ is known. It can be considered the "output" of our nanodevice and used as parameter $y_1$ within the Neural ODE. However, $y_1 = \Delta m_z$ results from complex magnetic configurations and dynamics that cannot all be determined experimentally. Additional parameters are necessary to describe this underlying dynamics, which may be represented by unknown internal variables: $\tilde{y}_2$ to $\tilde{y}_m$.

Here we develop a new scheme to train Neural ODEs in this context of real experiments where, in practice, the knowledge of the system is always limited. Our idea originates from the insight that it is possible to convert a set of first-order differential equations in

multiple variables into a single higher-order differential equation in one variable. For example, let us consider a two-dimensional ODE $[\dot{y}_1, \dot{\tilde{y}}_2] = [ay_1 + b\tilde{y}_2, cy_1 + d\tilde{y}_2]$ with a single hidden variable $\tilde{y}_2$. The equivalent form of this first-order system is a second-order ODE with variable $y_1$, $\ddot{y}_1 = (a + d)\dot{y}_1 + (bc - ad)y_1$, where $\tilde{y}_2$ no longer appears.

This simple derivation suggests that an appropriate way for training a Neural ODE of $m$ internal variables where only one variable $y_1$ is accessible is to train a Neural ODE where the state vector $\mathbf{y}$ is composed of $y_1$ and its $(m-1)$th-order derivatives (see Supplementary Note 8 for a discussion in arbitrary dimension). The drawback of using higher-order derivatives (Time-Derivative method) is the sensitivity to noise of derivatives, resulting in a relative noise level much larger than in the original signal (see Supplementary Note 8). To make the best use of the original information and to avoid any preprocessing procedures, in this work, we employ several successive time-delayed states as an alternative to derivatives: we consider the input vector

$$\mathbf{y}(t) = (y_1(t), y_1(t + \Delta t_d), y_1(t + 2\Delta t_d), \dots, y_1(t + (k-1)\Delta t_d)),$$
(1)

where $k$ is the dimension of the new vector and $\Delta t_d$ denotes a single delay time. Here we chose a positive $\Delta t_d$ value in our work. Using a negative $\Delta t_d$ value, as is usually done in the time-delay embedding literature, leads to equivalent results. These time-delayed variables contain all the information provided by the high-order derivatives, but are less prone to noise. This scheme, which we introduced here in a qualitative manner, can also be justified mathematically by using a formalism known as the

embedding theorem (see Supplementary Note 8). More specifically, theorems by Takens[43] and by Sauer et al.[44] state that if the sequence $\mathbf{y}(t)$ consists of scalar measurements of the state vector of a dynamical system, then under certain genericity assumptions, the time delay embedding provides a one-to-one image of the original set, provided $k$ is large enough. The prevalent application of employing delay embedding is to make short-term predictions of nonlinear time series[45,46]. The combination of Neural ODE with the delay embedding theorem enables making predictions for nonlinear time series with arbitrary lengths in a precise way, because the neural network provides a strong language to describe the system non-linearity and thus the physical pattern can be captured through training with a large number of observed data.

The use of delayed variables may seem equivalent to the use of time derivatives, as the latter is typically calculated by taking linear combinations of discrete samples of the data. However, as mentioned earlier, numerical derivatives amplify the noise present in the training data, making the Neural ODE training process much more difficult with time derivatives than with delayed variables. Supplementary note 3 discusses this issue in detail and provides an example comparing these two training techniques.

**Extension of the Neural ODEs formalism to deal with time-varying external inputs**. The second challenge for employing Neural ODEs for predicting the behavior of physical systems is to include time-varying external inputs, such as the anisotropy and the DMI changes in Fig. 1b–d. In this case, the time derivative of the $\mathbf{y}$ variable is not only dependent on its current state, but also related to the input at the current step, a situation that cannot be described in the traditional form of Neural ODE.

Supplementary Note 8 details how such inputs can be included in our approach. This note shows that, mathematically, an ODE system $\dot{\mathbf{y}} = f_\theta(\mathbf{y}, e(t), t)$ with the state vector $y$ and the input $e(t)$ of dimension $m$ can be converted into an $m^{\text{th}}$-order ODE in the first variable $y_1$, depending on $e(t)$ and the first to $(m-1)^{\text{th}}$-order derivatives of $e(t)$. Accordingly, a system with time-varying input can be modeled by augmenting the delay vector of Eq. (1) with the extra variables $(e(t), e(t+\Delta t_d), ..., e(t+(k-1)\Delta t_d))$, and used as input the $f_\theta$ function.

A system with multiple inputs can then be modeled by incorporating time-delayed versions of all inputs. As illustrated in Fig. 2a, b, we treat the time $t$ as an extended element of vector $\mathbf{y}(t)$ into the neural network and concatenate its time derivative, which is a constant one value, as a known output of the neural network . In this way, the external inputs at any moment can be chosen deterministically and given to the neural network. For a clearer visualization, the whole procedure of our technique is provided in Algorithm 1.

**Algorithm 1**. Training Neural ODEs using incomplete system dynamics and external input.

**input** : Time intervals $T = \{t_0, t_1, ..., t_{n-1}\}$ with uniformly spaced step $\Delta t$, time-dependent input $E = \{e(t_0), e(t_1), ..., e(t_{n-1})\}$, observed scalar output trajectory $Y = \{y(t_0), y(t_1), ..., y(t_{n-1})\}$, mini-batch time length $bt$, mini-batch size $bs$, iterations $N_i$, dimension of the new vector $k$ (number of delays $k-1$), a single time delay interval $\Delta t_d = \Delta t$, and Neural ODE parameters $\theta$ with forward function:
 **function** forward ($\mathbf{y}$):
 $\mathbf{t} \leftarrow \mathbf{y}[k]$ ▷ Extract the last dimension of vector $\mathbf{y}$.
 $\dot{\mathbf{y}} \leftarrow (f_\theta(\mathbf{y}[0:k-1], \mathbf{e(t)}), 1)$ ▷ The derivative of time $\mathbf{t}$ is constant $\mathbf{1}$.
 **return** $\dot{\mathbf{y}}$
**output** Updated $\theta$
**for** $iter = 1, ..., N_i$ **do**

(1) Randomly select mini-batch with the initial time $\mathbf{t}_b = \{t_{b_0}, t_{b_1}, ..., t_{b_{s-1}}\}$ ($b_i \in [0, n-bt]$, $i \in [0, bs-1]$, $i$ is an integer), mini-batch targets $\mathbf{y}_{\text{true}} = \{(y(\mathbf{t}_b), y(\mathbf{t}_{b+1}), ..., y(\mathbf{t}_{b+k-1}), {}_b), ..., (y(\mathbf{t}_{b+bt-1}), y(\mathbf{t}_{b+bt}), ..., y(\mathbf{t}_{b+bt+k-2}), \mathbf{t}_{b+bt-1})\}$, initial points $\mathbf{y}_0 = (y(\mathbf{t}_b), y(\mathbf{t}_{b+1}), ..., y(\mathbf{t}_{b+k-1}), \mathbf{t}_b)$, external input (at time step $\mathbf{t}_{b+i}$, $i \in [0, bt-1]$, $i$ is an integer) $\mathbf{e}(\mathbf{t}_{b+i}) = (e(\mathbf{t}_{b+i}), e(\mathbf{t}_{b+i+1}), ..., e(\mathbf{t}_{b+i+k-1}))$.

(2) Call the Neural ODE solver and compute the predicted output trajectory $\mathbf{y}_{\text{pred}}$ using current $\theta$.

(3) Update $\theta$ by taking an ADAM step on the mini-batch loss, which is defined as Mean Square Error (MSE) of $\mathbf{y}_{\text{pred}}$ compared to $\mathbf{y}_{\text{true}}$.

 **end**

**Application of Neural ODEs to predict the behavior of skyrmion-based systems**. We now test the validity of our approach with the single-skyrmion system of Fig. 1. We train a Neural ODE with dimension $k = 2$ by employing a three-layer neural network $f_\theta$, with 50 neurons in each hidden layer, using Algorithm 1 (see Methods) and the 50-nanosecond trajectory of Fig. 1d as training set. Figure 1e shows an outstanding agreement between the predicted training output of $\Delta m_z$ by Neural ODE and micromagnetic simulations. To evaluate the performance of the trained Neural ODE at extracting interesting physical quantities, we next use it to predict the intrinsic breathing frequency of the skyrmion system for different values of $K_u$ or $D$. In that case, the test inputs are composed of a pulse signal of $\Delta K_u$ (or $\Delta D$) and a constant value of $D$ (or $K_u$) to induce an oscillating magnetic response $\Delta m_z$ and thus to predict the corresponding frequency for specific material parameters $D$ and $K_u$. They are thus different from the sinusoidal waveforms of $\Delta D$ and $\Delta K_u$ used for training (Fig. 1d), which is important to test the ability of the neural network to generalize (see Methods section). The results, shown in Fig. 1f, g, again show excellent agreement with the predictions of micromagnetic simulations.

We then investigate the impact of the dimension of the Neural ODE on the prediction accuracy. Figure 2c compares the training process of a Neural ODE of dimension $k = 2$ with a Neural ODE of $k = 1$, i.e., without augmentation of delayed state (in this Figure, the anisotropy was used as sole input). The training error (mean square error, MSE) converges rapidly to zero for $k = 2$ but not for $k = 1$, for which it remains finite. The corresponding time-domain training outputs are shown in Supplementary Note 1. In general, we observed that in the absence of noise, a good model can be trained for any dimension $k \geq 2$. This result can be interpreted by the fact that the physical system, here a skyrmion, can essentially be described by two variables: the skyrmion radius and its phase[47]. For the modeling of noisy time series, a dimension of two is insufficient to train a good model, and higher accuracy can be obtained by increasing the number of delays. This result can be explained by the fact that gathering more information, i.e., adopting a delay vector of higher dimension, means less distortion of noise distribution and lower noise amplification when the time delay embedding is mapped into the original state space[48] (see Supplementary Note 2)

Further results regarding the training performance in terms of the number of neurons $N_h$ in the hidden layer, the sampling interval $\Delta t$ of the trajectory, the dimension $k$ of Neural ODEs, and different optimization algorithms can be found in Supplementary Note 1. Concerning the choice of $\Delta t_d$, this parameter should not be too small, as there would be almost no difference between different elements in a vector, and not too large, as the neighboring states may lose correlations. However, our results showed that the training results do not depend too significantly on $\Delta t_d$, and therefore, we took $\Delta t_d$ equal to the sampling interval.

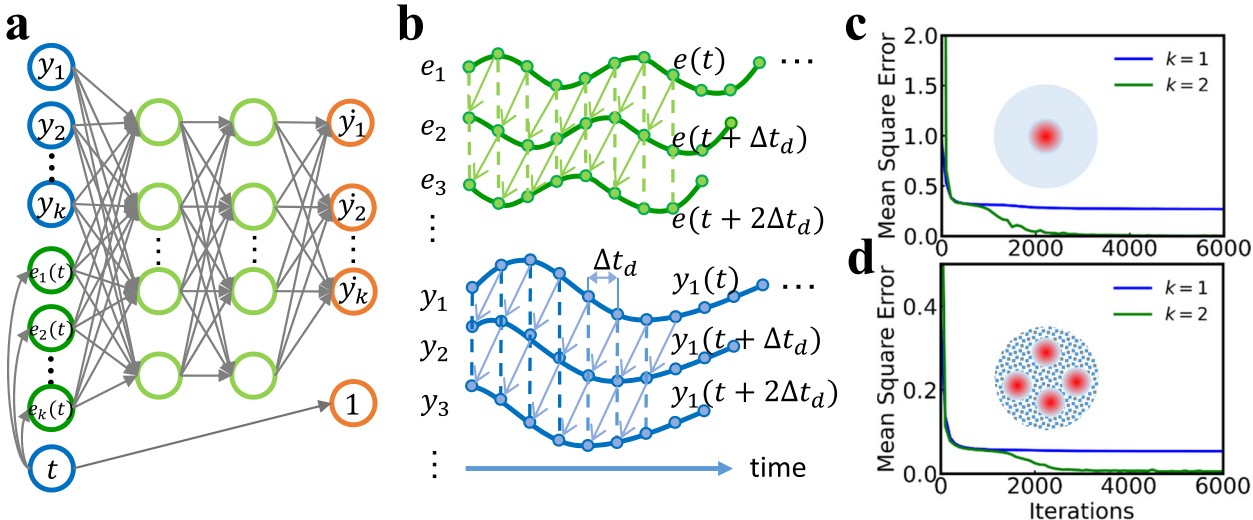

**Fig. 2 Extending the Neural ODE formalism to predict spintronic results.** A wide range of dynamical systems can be modeled using ODEs, such as the simple pendulum motion, skyrmion-based devices, and spintronic oscillator dynamics. However, in real-world applications, the underlying physical dynamics are not always fully measurable, or accessible, which means that the dynamics of some hidden parameters in ODEs models are unknown. Our goal is to model a Neural ODE, $\dot{\mathbf{y}} = f_\theta(\mathbf{y}, e(t), t)$, where $f$ is defined by a neural network and $e(t)$ is the time-dependent input into the system, using the incomplete information of the system dynamics. **a** Schematic graph of the neural network ($f_\theta$) in a Neural ODE. The input to the neural network consists of two parts, one is the $k$ dimensional vector related to the observed system dynamics, where $y_1$ is the observed dynamics and $y_2$ to $y_k$ are the time-delayed dynamics of $y_1$ (as shown in **b**), another part comprises the time-dependent external inputs, in which $e_1$ is the original input and $e_2$ to $e_k$ are the time-delayed versions of $e_1$. The output of the neural network is a vector of time derivatives of the corresponding input system dynamics ($y_1, y_2, ..., y_k, t$). Here, the derivative of the time variable $t$ is 1, which is determined as a prior knowledge. **b** Illustration of the time-domain system dynamics and external time-dependent input dynamics used for modeling the Neural ODEs. The blue curves are the system dynamics, where $y_1$ is the original observed trajectory, $y_2 = y_1(t + \Delta t_d)$ is one time step shifted of $y_1$, $y_3 = y_1(t + 2\Delta t_d)$ is two time steps shifted of $y_1$, etc. Here, $\Delta t_d$ denotes the single time delay interval. The green curves are the external inputs, where $e_1 = e(t)$ is the original input dynamics, $e_2 = e(t + \Delta t_d)$ is one time shifted of $e_1$, $e_3 = e(t + 2\Delta t_d)$ is two time steps shifted of $e_1$, etc. Through the augmentation, the reconstructed system dynamics ($y_1, y_2, y_3, ..., y_k$) containing the information of the unknown state variables can be used to train the Neural ODEs and then the trained Neural ODEs can be applied to make predictions for other inputs. **c**, **d** Training error (Mean Square error, MSE) as a function of iterations for $k = 1$ and $k = 2$ for a one-skyrmion system (**c**) and a multi-skyrmions system with grain inhomogeneity (**d**) with electric voltage as input through the VCMA effect.

We also validated our algorithm in more complex situations. Neural ODEs were able to predict the behavior of multi-skyrmions systems with grain inhomogeneities exhibiting a distribution of perpendicular magnetic anisotropy (PMA) $K_u$, and with voltage as input, (see Fig. 2d and Supplementary Fig. 1c). In this case, the skyrmions show coherent oscillations, i.e., all skyrmions oscillate in phase with the same frequency[49], and grain inhomogeneity mainly distorts the shapes of skyrmions. Therefore, the averaged magnetization dynamics can be described in the same way as the idealized one-skyrmion system, for any dimension $k \geq 2$. Neural ODEs also worked when electric current is used as an input, causing the skyrmion to rotate within the device (see Supplementary Note 5).

To summarize, skyrmions systems are usually modeled by time-consuming micromagnetic simulations, and developing faster models is a challenging task. Isolated skyrmion dynamics is often modeled by analytical equations that neglect the skyrmion deformation in confined systems. On the other hand, it is particularly difficult to model multi-skyrmions system, especially taking into account imperfections at the material level. Here, we showed that Neural ODEs can be trained to model these different situations.

**Benchmark test for reservoir computing: Mackey-Glass time series prediction**. We now show that the Neural ODEs trained in

the previous section can be used without any change of parameters to predict the response of the spintronic system in a different setting, and with inputs that vary in a very different way, with computation time considerably reduced compared to micromagnetic simulations. We focus on a neuromorphic task called reservoir computing that exploits the intrinsic memory of complex dynamical systems, and apply it to the case of reservoirs made of single and multiple skyrmion textures[50–52]. The reservoir input corresponds to a chaotic time series (Mackey-Glass chaotic series, see Methods section), and the goal of the task is to predict the next steps in the time series (Fig. 3a). The response of spintronic devices to such time series is particularly long to simulate with micromagnetic simulations. We simulated a reservoir computing experiment using the Neural ODEs trained in the previous section (which required 20 min of simulation time), as well as using micromagnetic simulations, as a control (requiring four days of simulation time). Figure 3b, c show the time series predicted by a one-skyrmion system modeled by micromagnetic simulations and Neural ODEs, respectively, in comparison with the true trajectory (blue) of the Mackey-Glass time series. This data is presented in a situation where the skyrmion reservoir has to predict the next value in the Mackey-Glass time series ($H = 1$), and in a situation where it has to predict the value happening 25 steps later ($H = 25$), a much more difficult task due to the chaotic nature of the Mackey-Glass time series. In the

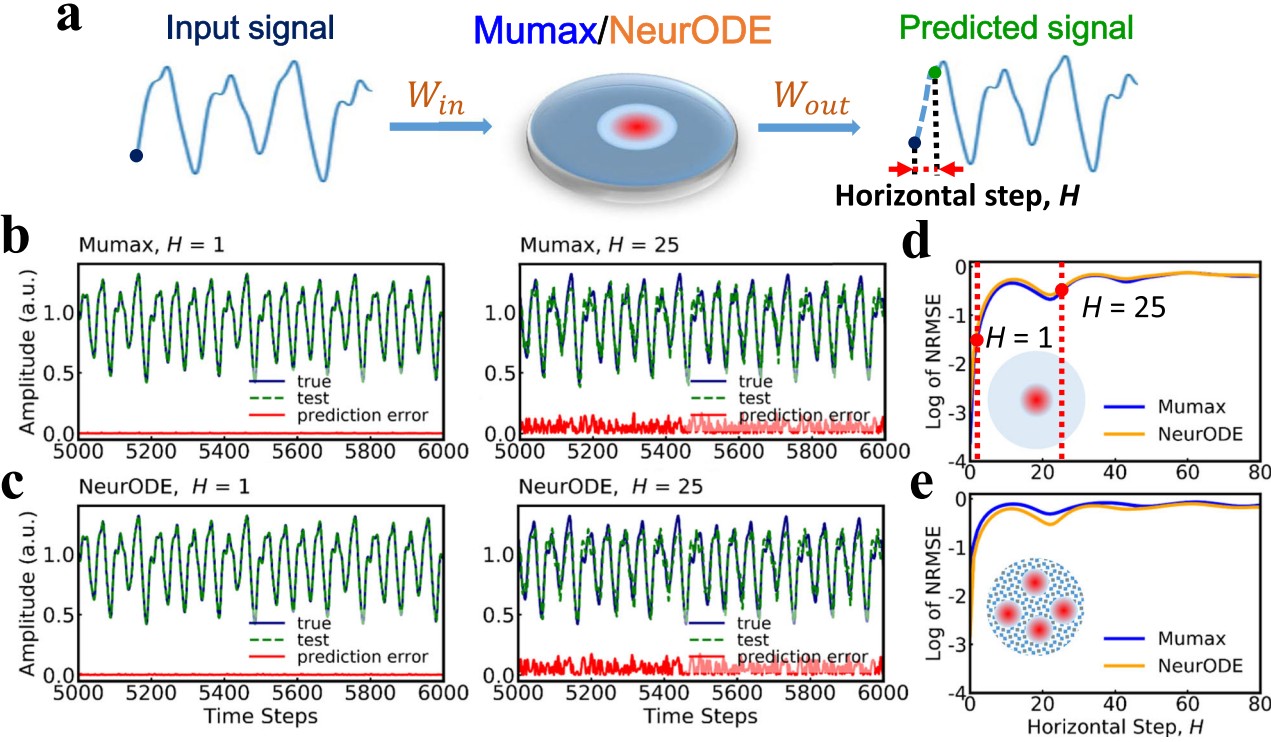

**Fig. 3 Modeling of a sophisticated spintronic task, Mackey-Glass time series prediction with skyrmion systems, using Neural ODEs. a** Schematic graph of the procedure for doing the prediction task. The purpose is to predict the Mackey-Glass time series at a future time. The input signal, preprocessed through a read-in matrix $W_{in}$, is fed into the reservoir, which is a skyrmion system modeled by the trained Neural ODE or micromagnetic simulations. A trained output matrix $W_{out}$ is used for reading out the reservoir states and providing the predicted signal. **b**, **c** Selected testing (green dashed) results for prediction horizontal step $H = 1$ (for short-term prediction) and $H = 25$ (for long-term prediction), predicted by the one-skyrmion system modeled by micromagnetic simulations in **b** and the Neural ODE in **c**, in comparison with the true trajectory (blue) of Mackey-Glass time series. The red curves show the prediction error compared to the true trajectory. **d**, **e** Normalized root mean square error (NRMSE) as a function of prediction horizontal step $H$, in log scale, for the testing set by using the trained Neural ODE (orange) and micromagnetic simulations (blue) for the one-skyrmion system in **d** and the multi-skyrmions system in **e**.

$H = 1$ situation, the predictions of micromagnetic simulations and the Neural ODE match the true series perfectly, while for $H = 25$, a small prediction error happens, which appears consistent in both cases. To verify if the Neural ODE and micromagnetic simulations give equivalent predictions, Fig. 3d presents the accuracy of the prediction of the Mackey-Glass series, expressed in terms of Normalized Root Mean Square Error (NRMSE) as a function of horizontal prediction step $H$ (a lower value of the NRMSE means a more accurate prediction). We see that the NRMSE computed by the Neural ODE matches the one from micromagnetic simulations very precisely. Figure 3e presents the same result for a multi-skyrmions system, where once again, the results of the Neural ODE match those of micromagnetic simulations accurately. We also evaluated the prediction performance in terms of the number of virtual nodes of the reservoir, the time duration for each preprocessed input staying in the reservoir, and other techniques of training the output weights (see Supplementary Note 4).

This demonstration of a demanding benchmark task exploiting the details of skyrmion dynamics indicates the potential of skyrmion system for reservoir computing, and highlights the quality of predictions by the Neural ODEs, with considerable improvement in computational efficiency (the Neural ODEs simulation were 200 times and 360 times faster than the micromagnetic simulations for the one-skyrmion system and multi-skyrmions system).

**Predicting the experimental measurements of a nanoscale spintronic oscillator.** We now apply our approach to modeling real experimental data, obtained using the setup of ref. [17]. This work showed experimentally that a nanoscale spintronic oscillator can be used as a reservoir computer to achieve a spoken digit recognition task (based on a principle similar to what we implemented in the skyrmion system). In this regime, the nanoscillator is functioning as a nonlinear node to map the input signal into a higher dimensional space in which the input can be linearly separable (see Fig. 4a).

Modeling this experiment is a difficult challenge. Until now, analytical models of spintronic oscillators could reproduce experiments only qualitatively: it is challenging to construct a reliable model due to the high non-linearity of the devices as well as the impact of noise appearing in the experiments. Here, we firstly train a Neural ODE model of the oscillator dynamics by using only 5 ms of experimental data (see Methods section and Fig. 4), then we use the trained model to predict the whole spoken digit recognition experiment of Torrejon et al.[17] (Fig. 5). The results reported in Figs. 4 and. 5 use cochlear preprocessing (see Methods section).

Figure 4 b shows the 5-ms trajectories used for training, as well as the result of trained Neural ODE of dimension $k = 2$, showing remarkable agreement. For a more quantitative assessment, Fig. 4c shows the training loss (MSE) of Neural ODEs with dimension $k$ ranging from one to four (a smaller MSE means a higher accuracy). The models with dimension two or greater reach a much smaller loss than a model of dimension one. This result is consistent with the conventional modeling of these devices through coupled amplitude and phase equations, requiring therefore at least a two-dimension ODE[18]. It is also remarkable that the losses can be extremely close to zero, but not arbitrarily close. The impossibility of reaching zero loss can be

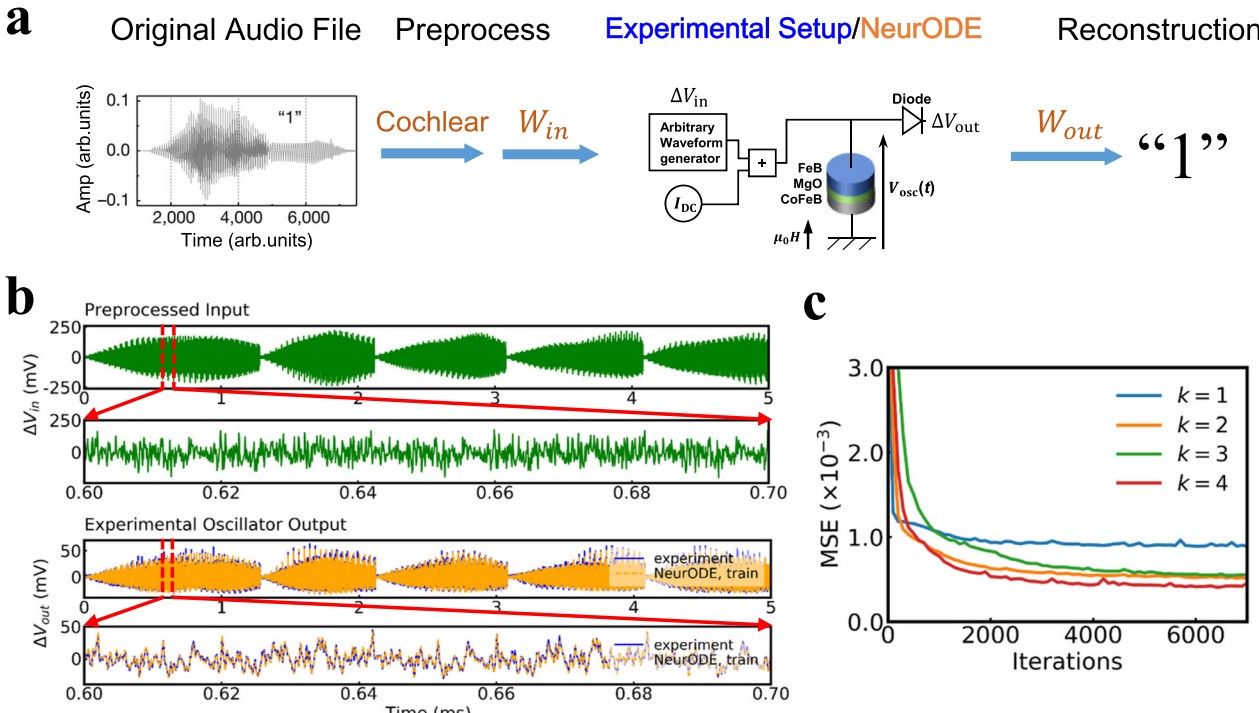

**Fig. 4 Prediction of experimental results using Neural ODEs (train results). a** Principle of the experiment. The original spoken digit in the audio waveform is preprocessed, by cochlear or spectrogram filtering, to form the preprocessed input into the oscillator. The output of the digit is reconstructed by reading out the recorded oscillator output through a trained matrix $W_{out}$ (see Methods section). The purpose of modeling is to predict the experimental oscillator output given any preprocessed input $\Delta V_{in}$. This Figure shows the results obtained using cochlear filtering (results using spectrogram filtering are reported in Suppl. Fig. 8). **b** Training output trajectory of voltage $\Delta V_{out}$ predicted by Neural ODE (dashed orange) with corresponding preprocessed input $\Delta V_{in}$, in comparison with the experimental measurement (blue) for $k = 2$. A training set of 5-ms dynamics is adopted from the first utterance of the first speaker. A three-layer neural network $f_\theta$ with each hidden layer of 100 units is trained. **c** Training loss (MSE) of Neural ODE with $k = 1, 2, 3, 4$ as a function of iterations.

attributed to the existence of noise in the experimental data, whereas the Neural ODE is entirely deterministic.

Next, the trained model can be utilized to predict the results of the spoken digit recognition experiment. Figure 5 reports results obtained using the Neural ODE of dimension $k = 2$. The aim of the task, described in Methods, is to classify the ten digits spoken by five different female speakers. Realizing this task experimentally involves a week-long experimental campaign, while it can be simulated in two hours using a trained Neural ODE. This task can also not be simulated by micromagnetic simulations, as it would require 716 years of simulation time on our reference GPU (this number was extrapolated based on the simulation of the dynamics of a nano-oscillator during one microsecond). Selected response output trajectories predicted by the trained Neural ODE and corresponding experimental output are shown in Fig. 5a, showing very good agreement.

We can now see how these results translate in terms of spoken digits recognition rate. Figure 5c shows the recognition rate on spoken digits as a function of the number of utterances used. We see that the results obtained by a Neural ODE without noise do not match those of experiments. Incorporating noise within the Neural ODE is essential to predict the experimental data (this is particularly the case here, as reservoir computing is very susceptible to the disturbance of noise). To do this, we can rely on the 5-ms training data. We extract the error distribution by computing the difference between the output trajectory predicted by Neural ODE and the experimental measurement and fit this error to a Gaussian law (Fig. 5b). We then inject Gaussian noise in the Neural ODE, as an additional input, with an amplitude

chosen so that the output noise ($\sigma_{out}$) of the Neural ODE matches the standard deviation of the data ($\sigma_{err}$) of Fig. 5b (see Methods section). When using the Neural ODE augmented with noise to simulate the spoken digits experiments, the digit recognition rates now match the experimental data very closely (Fig. 5c), making the Neural ODE augmented with noise a powerful tool to predict long and complex spintronic experiments.

Interestingly, the noise plays a role of suppressing the overfitting of the output states from reservoir, actually improving the recognition rate. Conversely, we saw that task performance deteriorates with the injection of noise if the data has been preprocessed with the spectrogram filtering method, indicating that the output states from reservoir is under-fitted. This difference arises because the preprocessing procedure also contributes to the nonlinear transformation of input signal, and the cochlear method of preprocessing provides more nonlinearity than that of the spectrogram method[17] (more information about the impact of noise, and in particular in the spectrogram situation, is provided in Supplementary Note 7). It is remarkable that the Neural ODE augmented with noise is able to predict so subtle behaviors, which we were not able to realize from experimental data only.

## Discussion and related works

Our approach allows learning the underlying dynamics of a physical system from time-dependent data samples. Many works today seek to use deep neural networks to predict results in physics. They are used to find abstract data representations[20], recover unknown physical parameters[22], or discover the specific

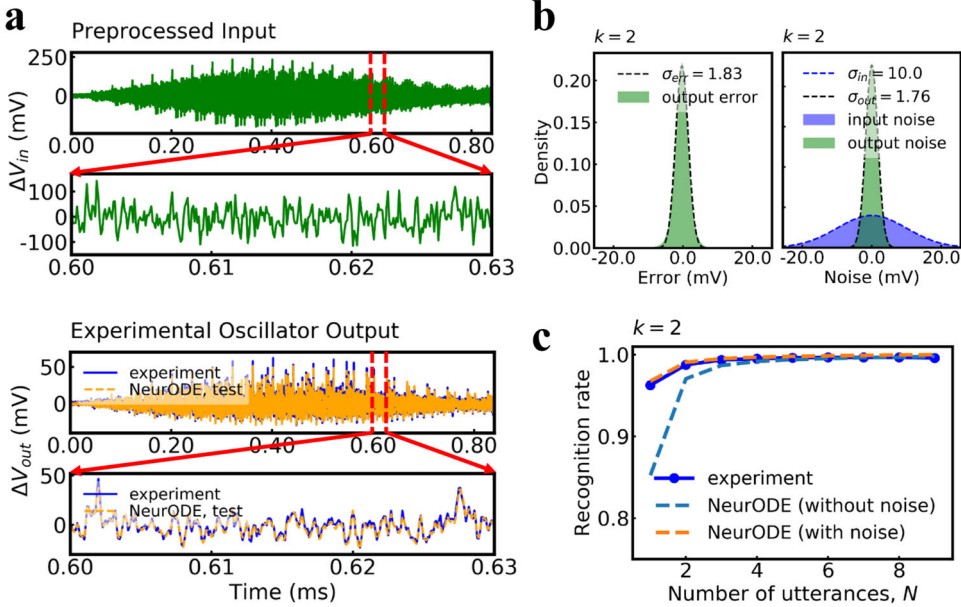

**Fig. 5 Prediction of experimental results using Neural ODEs (test results). a** Selected response output signals predicted by the trained Neural ODE with corresponding preprocessed input, in comparison with the experimental output for digit eight of the third utterance of the third speaker. **b** Left to right: error distribution (green shadow) and fitted Gaussian distribution (dashed curve) extracted by computing the difference between the output predicted by Neural ODE and the experimental measurement, a Gaussian noise (purple shadow) added in the preprocessed input into the Neural ODE and the corresponding noise distribution (green shadow) with fitted probability density function (pdf) (black dashed curve) in the predicted output trajectory solved by Neural ODE. $\sigma_{in}$ was adjusted so that $\sigma_{out} = 1.76$ mV is close to ($\approx$) $\sigma_{err} = 1.83$ mV. **c** Spoken digit recognition rate in the testing set as a function of utterances $N$ used for training. Because there are many ways to pick the $N$ utterances, the recognition rate is an average over all $10!/[(10 - N)!N!]$ combinations of $N$ utterances out of the 10 in the dataset. The solid curve, blue dashed curve, orange dashed curve is the experimental result, Neural ODE result with noise considered, Neural ODE result without any noise, respectively.

terms of functions[23–25]. Other research uses recurrent neural network-based models[26–28] to learn and make predictions. These methods usually incorporate prior knowledge on the physical system under consideration, such as molecular dynamics[24,26], quantum mechanics[20], geospatial statistics[28], or kinematics[25] to help their models train faster or generalize. Few of these discrete models manage to include the relevant driving series to make predictions. Neural ODEs hold many advantages over the conventional neural networks used in these works: backpropagation occurs naturally by solving a second, augmented ODE backward in time; stability is improved with the use of adaptive numerical integration methods for ODEs; constant memory cost can be achieved by not storing any intermediate quantities of the forward pass; continuously-defined dynamics can naturally incorporate data which arrives at arbitrary times. However, until our work, two challenges remained to apply Neural ODEs to the prediction of the behavior of physical systems: the impossibility, in most practical cases, to acquire the dynamics of the set of state variables of the system, but also the need to take into account the external inputs that affect their dynamics.

Our work addresses both issues, and before ours, other works have attempted to solve the first issue. One way is to introduce the inductive bias via the choice of computation graphs in a neural network[25,53–59]. For example, by incorporating the prior knowledge of Hamiltonian mechanics[58,59] or Lagrangian Mechanics[55–57] into a deep learning framework, it is possible to train models that learn and respect exact conservation laws. These models were usually evaluated on systems where the conservation of energy is important. Similarly, another strategy to deal with a dataset with incomplete information is through augmentation of original dynamics[60,61]: extensions of Neural ODEs at the second-order[60] or higher-order[61], can learn the low-dimensional physical dynamics of the original system. However, nearly all the proposals mentioned above require the knowledge of

additional dynamical information, such as higher-order derivatives, or extra processing of the original low-dimensional dynamics, which is not appropriate for dealing with noisy time series. Neural ODE integrated with external inputs has also been studied in some previous literature[62–64]. Augmented Neural ODEs[63] solve the initial value problem in a higher-dimensional space, by concatenating each data point with a vector of zeros to lift points into the additional dimensions. This strategy avoids trajectories intersecting each other, and thus allows modeling more complex functions using simpler flows, while achieving lower losses, reducing computational cost, and improving stability, and generalization. Parameterized Neural ODE[64] extends Neural ODEs to have a set of input parameters that specify the dynamics of the Neural ODEs model such that the dynamics of each trajectory are characterized by the input parameter instance.

We emphasize here that our idea is closely related to the classical theorem of time delay embedding for state space reconstruction, where the past and future of a time series containing the information about unobserved state variables can be used to define a state at the present time. The theorem was widely applied for forecasting in many real-world engineering problems[45,46,65], but was largely restricted into making very short-term predictions for lack of modeling frame specifically designed for time series. The advent of Neural ODEs, whose formalism naturally incorporates time series, allows predictions of arbitrary length and high accuracy to be made by training a system equivalent to the original physical system. Additionally, until our work, Neural ODEs-based methods for modeling time series had only been tested in a few classical physical systems, such as the ideal Mass-Spring system, Pendulum and Harmonic oscillators. Our work is the first one to apply Neural ODEs to predict the behavior of nanodevices, by resolving the above issues.

Our method also provides a significant improvement in time efficiency compared to conventional simulation platforms. For

example, the Mackey-Glass prediction task with a reservoir of skyrmions takes only 20 min for the trained neural ODE, while the micromagnetic simulations need 3 days (5 days) to run it on the one-skyrmion system (multi-skyrmion system). To model the dynamics of the spintronic oscillator using real experimental data, output time series of 5-ms duration are sufficient to train a complete Neural ODE model, capable of predicting the system dynamics with any input. The Neural ODE simulation time does not dependent directly on the size of the system, but only on the number of dimensions and the number of data points to be predicted; the possibility to train a Neural ODE is therefore mostly determined by the availability of appropriate training data. Therefore, constructing a reliable and accurate model is not the only purpose of Neural ODEs, they can be a strong support for fast evaluation, verification, and optimization of experiments. In this way, our work also paves a way to Neural ODE-assisted or machine learning-integrated simulation platform development.

Last but not least, Neural ODEs can be used for modeling systems featuring different types of behaviors, provided that examples of the different behaviors were included in the training dataset. Supplementary Note 6 shows an example, where a single Neural ODE can model a device, which, depending on the value of the external magnetic field, exhibits either a switching or a sustained oscillatory behavior. A limitation of Neural ODE, however, is that they cannot be trained to model systems exhibiting profoundly stochastic behavior, as is sometimes observed in room-temperature switching of magnetic tunnel junction[66] or domain wall motion in some regimes[67]. Neural ODEs are adapted for systems obeying deterministic equations. Future work regarding the modeling of stochastic behaviors of a physical system using Neural ODE remains to be explored, which could rely on recent developments of stochastic Neural ODE theory[68,69].

In conclusion, we have presented an efficient modeling approach for physical ODE-based systems, and highlighted its excellent performance in modeling real-world physical dynamics. The training data can be a single observed variable, even if the system features higher-dimensional dynamics. We have shown that the method can not only be applied to model ideal data from simulations, but that it is also remarkably accurate for modeling real experimental measurements including noise. The trained model shows a remarkable improvement (hundreds of times faster) in computational efficiency compared to standard micromagnetic simulations. We have shown that Neural ODE is a strong support for making experimental predictions and dealing with complex computation tasks, such as the task of Mackey-Glass time-series predictions and spoken digit recognition in reservoir computing. In particular, we demonstrate its use in modeling complex physical processes in the field of spintronics, which is considered one of the most promising future technologies for memory and computing. The proposed method is a promising tool for bridging the gap between modern machine learning and traditional research methods in experimental physics, and could be applied to a variety of physical systems.

## Methods

**Micromagnetic simulations**. Our micromagnetic simulations are performed in the MuMax3 platform (abbreviated to Mumax in the main text)[70], an open-source GPU-accelerated micromagnetic simulation program. The default mesh size of $1 \text{ nm} \times 1 \text{ nm} \times 1 \text{ nm}$ is used in our simulations. The following material parameters are adopted: exchange stiffness $A = 15$pJ/m, saturation magnetization $M_s = 580$ kA/m, damping constant $\alpha = 0.01$, interfacial DMI strength $D = 3.5$mJ/m$^2$, and default PMA constant of the ferromagnetic layer $K_u = 0.8$ MJ/m$^3$. In addition, we set the VCMA coefficient $\xi$ as 100 fJ · V$^{-1}$m$^{-1}$ based on some recent experiments[71,72]. Here, the typical thickness of the insulating layer is 1 nm. Under these conditions, with an applied voltage of 0.1 V (an electric field of 0.1 V/nm), the PMA constant in the ferromagnetic layer will change by 10 kJ/m$^3$.

For the simulation of single-skyrmion dynamics with voltage input (system used in Figs. 2 and 3), a nanodisk with a diameter of 80 nm is used. For the training set, the external input voltage to the system is random sine voltage with a frequency of 4 GHz and with amplitude ranging from −2 V to 2 V (corresponding to variation of PMA value $\Delta K_u$ from −0.2 to 0.2 MJ/m$^3$, see Suppl. Fig 1). For the multi-skyrmions system with grain inhomogeneity, the diameter of the nanodisk is 120 nm, and the grain size is 10 nm. Random 20% PMA variation, random 20% DMI strength variation, and 5% random cubic anisotropy direction variation are applied. The external input voltage to the system is a random sine voltage with a frequency of 4 GHz and with amplitude ranging from −2 V to 2 V (corresponding to variation of the PMA value $\Delta K_u$ from −0.2 to 0.2 MJ/m$^3$). For the testing set of Mackey-Glass prediction task (results in Fig. 3), the input is a time varying $\Delta K_u$ in the form of preprocessed MG time series with a time interval of 10 ps (as shown in Suppl. Fig 5).

For the parameters-based simulations in Fig. 1, the diameter of the nanodisk is 100 nm. For the training set, the external input $\Delta K_u$ is a random sine with a frequency of 4 GHz and with amplitude ranging from −0.05 to 0.05 MJ/m$^3$, fluctuated ~0.8 MJ/m$^3$. The external input $\Delta D$ is a random sine with a frequency of 0.4 GHz and with amplitude ranging from −0.4 to 0.4 mJ/m$^2$, fluctuated around 3.0mJ/m$^2$ (see Fig. 1d). The perpendicular average magnetization variation $\Delta m_z$ of the system is recorded every $p = 2.5$ ps as output. For the testing set, to get the response frequency of each material value of $K_u$ ($D$), we firstly supply a pulse with amplitude $\Delta K_u = 0.04$MJ/m$^3$ ($\Delta D = 0.1$mJ/m$^2$) lasting for 1 ns, then the magnetization variation $\Delta m_z$ is recorded. Finally, a Fourier transform is conducted on the output trajectory of $\Delta m_z$ to obtain the frequency. Simulation time of 37 mins, 41 mins, and 43 mins for 50 ns dynamics are needed for the training set of the one skyrmion system, multi-skyrmions system, and parameters-based system simulations.

**Training method of Neural ODE**. To train the Neural ODE, we build a single-trajectory training set $\mathbf{y}_{\text{true}}$ consisting of $n$ data points sampled from the output trajectory $\Delta m_z$ for the skyrmion system and from $\Delta V_{\text{out}}$ for the experimental oscillator with a time interval $\Delta t$. We use the mean squared error (MSE) between these points and the corresponding trajectories predicted by the Neural ODE $\mathbf{y}_{\text{pred}}$ over all time steps as the "loss function", i.e., the value that the training process aims at minimizing. To achieve the minimization of the loss, the gradients of the loss with respect to the parameters $\theta$ are computed through a technique called adjoint sensitivity method[39], then the $\theta$ parameters can be updated by using gradient descent optimization algorithms (usually stochastic gradient descent or Adaptive Moment Estimation[73]), until the MSE approaches zero. In this work, we train a Neural ODE in the form of $\dot{\mathbf{y}} = f_\theta(\mathbf{y}, \mathbf{e}(t), t)$ in which $\mathbf{y}(t) = (y_1(t), y_1(t + \Delta t_d), y_1(t + 2\Delta t_d), \ldots, y_1(t + (k-1)\Delta t_d))$ and $\mathbf{e}(t) = (e(t), e(t + \Delta t_d), e(t + 2\Delta t_d), \ldots, e(t + (k-1)\Delta t_d))$ with Adaptive Moment Estimation.

The number of training data points $n = 10,000$, 15,000, 10,000, and 50,000 and validation data points of 5000, 5000, 5000, 10,000 are used for the one skyrmion system, the multi-skyrmions system, the parameter-based system and experimental data of oscillator, respectively. A sampling interval $\Delta t$ for training is determined according to the original recorded output period $p$. Specifically, $p = 2.5$ ps, 100 ns are used for Mumax simulations and experimental measurements, respectively. In principle, a Neural ODE system can be properly modeled as long as the training data are continuous in time. Considering the trade-off between the accuracy of the model and the time efficiency for processing, $\Delta t = 2p$ is chosen for modeling the simulation data from modeling of data and $\Delta t = p$ for modeling the experimental data of oscillator (see training results with $\Delta t = p, 2p, 4p, 5p$ in Supplementary Note 1). The $f_\theta$ function of the Neural ODEs is a three-layer feedforward neural network. Each hidden layer features 50 units for Mumax data modeling, and 100 units for modeling of experimental data. The activation function is the tanh function except for the output layer. The values of the weights are initialized from the standard normal distribution with a mean of 0 and a standard deviation of 0.1. To train the Neural ODE using the Algorithm 1, we set the mini-batch time length $bt = 20$ and mini-batch size $bs = 50$. We use the optimization algorithm Adaptive Moment Estimation (Adam) with a learning rate 0.001 to update the hidden weight with loss gradients of MSE. Fixed step fourth-order Runge-Kutta method with 3/8 rule is used as the Neural ODE solver.

During the training, time intervals $T$ are normalized by a multiplying a factor of $\delta = 0.0125/p$ as default. In addition, normalized input and output is used for training. For the skyrmion-based model, the normalized input is shown in Fig. 1d: the variation of magnetization output $\Delta m_z$ is multiplied by ten for training. For modeling the experimental data of oscillator, both $\Delta V_{\text{in}}$ and $\Delta V_{\text{out}}$ are multiplied by ten for training.

The prediction is made by specifying an initial value of $\mathbf{y}$ and applying the time-varying inputs $\mathbf{e}(t)$ into the trained Neural ODE. The test set is used for the evaluation of the prediction performance of the trained Neural ODE. The total number of testing points is $n = 800$ for each of the 37 different values of $D$ and $K_u$ in Fig. 1, $n = 500,000$ for the Mackey-Glass time series, $n = 522,8800$ for the experimental oscillator with the cochlear method, and $n = 850,7200$ for the experimental oscillator with the spectrogram method.

**General concept of reservoir computing**. Reservoir computing is a computational framework derived from recurrent neural network models and suited for

temporal/sequential data processing[74]. A reservoir is a network of interconnected nonlinear nodes with feedback. It transforms non-linearly its input signal into a higher-dimensional space, so that the resulting signal can be linearly separable through a simple readout to a desired output. The key benefit behind reservoir computing is that only the output layer from the reservoir states is trained using a simple linear regression mechanism, such as ridge regression, as all connections inside the reservoir are kept fixed. Reservoir computing thus brings great advantage for hardware implementation, because the computational power of naturally available systems, such as a variety of physical systems, substrates, and devices, can be leveraged for such computation tasks.

In general, there are several requirements for efficient computing by a physical reservoir. First, high dimensionality ensures the mapping from inputs signal into a high-dimensional space through a nonlinear transformation, so that the originally inseparable inputs can be separated in classification tasks, and the spatiotemporal dependencies of inputs can be extracted in prediction tasks. Second, the fading memory (or short-term memory) property is necessary so that the reservoir state is dependent on the current inputs and recent past inputs. Such a property is particularly important for processing temporal sequential data in which the history of the states is essential.

In the Mackey-Glass prediction task, a skyrmion system can emulate a reservoir when it is in transient dynamics under an external input of a varying current or voltage. Similarly, in the spoken digit recognition task, a single nonlinear oscillator is used as a reservoir by leveraging its current-induced transient response. In both cases, an additional preprocessing step, where the original input is multiplied by a mask, usually a random matrix, is needed to enable the virtual nodes to be interconnected in time. The output states from the reservoir are reconstructed in a similar way, where the states from the virtual nodes are read consecutively in time.

**Mackey-Glass prediction task.** In a chaotic system, small perturbations can result in radically different outcomes. The prediction of a chaotic system is thus a problematic task. As a chaotic system, the Mackey-Glass equation is generated from a delay differential equation (DDE),

$$\frac{dx(t)}{dt} = \frac{\beta x(t-\tau)}{1 + x^{10}(t-\tau)} - \gamma x(t), \tag{2}$$

where $x(t)$ is a dynamical variable, $\beta$ and $\gamma$ are constants. Chaotic time series can be achieved with $\beta = 0.2$, $\gamma = 0.1$ and $\tau = 17$[75].

In the main paper, our goal is to predict the Mackey-Glass time series at a future time step: the preprocessed input signal at the current time is fed into the reservoir, which maps it non-linearly into higher-dimensional computational spaces (see Fig. 3a), and a trained output matrix $W_{out}$ is used for reading out the reservoir states. The number of steps between the future time step and the current step is defined as prediction horizontal step $H$. $W_{out}$ is different for the different $H$ values.

More precisely, the dataset for the prediction task is prepared in the following way. First, Eq.(2) is solved for 100,000 integration time steps with dt = 0.1. Before the data are processed by the reservoir, they are downsampled with a downsampling rate of 10 to remove the possible redundancy in the input data[75]. Thus, we obtain 10,000 data points in total. The first $5000 + H$ data points are used for the training, and the rest $5000 - H$ are for testing. The first stage of the masking procedure is a matrix multiplication $W_{in} \cdot M_o$, where $W_{in} \in \mathbf{R}^{N_r \times 1}$ is the mask matrix with data values drawn from a standard normal distribution and $M_o \in \mathbf{R}^{1 \times L}$ is the original input data. Here $L = 5000 + H$ is the number of the scalar input data points and $N_r$ is the reservoir size. We adopt $N_r = 50$ in the main text. As a consequence of the masking, we obtain the data matrix $M_e = W_{in} \cdot M_o \in \mathbf{R}^{N_r \times L}$. Then $M_e$ is column-wise flattened into a vector $e \in \mathbf{R}^{L \cdot N_r}$ and then fed into the reservoir of skyrmion systems.

Each of the value from $e$, multiplied by a voltage of 1.6 V ($\Delta K_u = 0.16$ MJ/m$^3$) for the one skyrmion system and a voltage of 1 V ($\Delta K_u = 0.1$ MJ/m$^3$) for the multi-skyrmions system, is provided as preprocessed input into the reservoir (modeled by a trained Neural ODE or Mumax) for $t_{step} = 4p = 10$ ps to make sure there is an effect of the input on the reservoir dynamics. In the following, the reservoir dynamics is recorded for every $t_{step}$ to form a vector of $M_y \in \mathbf{R}^{L \cdot N_r}$, which is then unflattened into a response matrix $M_x \in \mathbf{R}^{N_r \times L}$ for output reconstruction. We use the matrix $A \in \mathbf{R}^{N_r \times T_r}$ consisting the first $T_r = L - H$ columns of $M_x$ for training the read out Matrix. The teaching matrix $B \in \mathbf{R}^{1 \times T_r}$ consisting the last $T_r$ of the original signal $M_o$ is the time series to be predicted. A read out matrix $W_{out}$ is therefore constructed through the method of ridge regression,

$$W_{out} = (A \cdot A^T + \mu I)^{-1} (A \cdot B^T), \tag{3}$$

where $\mu = 10^{-4}$ is used as regularization parameter. To evaluate the performance of the trained matrix $W_{out}$, NRMSE is calculated on the prediction results of the testing set $y_{pre}$ compared to the true trajectory of MG series $y_{tar}$,

$$NRMSE = \sqrt{\frac{1}{n_s \sigma_{tar}^2} \sum_{i=0}^{n_s} (y_{tar}(i) - y_{pre}(i))^2}. \tag{4}$$

**Spoken digit recognition task.** In the task of spoken digits recognition, the inputs are taken from the NIST TI-46 data corpus. The input consists of isolated spoken digits said by five different female speakers. Each speaker pronounces each digit ten times. The original input signals of the spoken digits are preprocessed using two

different filtering methods: spectrogram and cochlear models. In both methods, firstly, each word is broken into $N_\tau$ time intervals of duration $\tau$. Here, $N_\tau$ can be different for different speakers. Then in each interval $\tau$, a frequency transformation is performed to convert the signal into the frequency domain with $N_f$ channels. This frequency signal is then processed by multiplying a mask matrix $W_{in}$ containing $N_f \times N_\theta$ random binary values for each interval to obtain $N_\theta \times N_f$ values in total as input to the oscillator. The number of virtual neurons is $N_\theta = 400$. Each preprocessed input value is consecutively applied to the oscillator as a constant current for a time interval $\theta = 100$ ns. For the classification task, the response matrix $S$ consisting of the output of all neuron responses for all of the $N_\tau$ intervals from $N$ utterances of ten digits of five speakers is used for training. The target matrix $Y$ contains the targets for each interval, which is a vector of 10 with the appropriate digit equaling to 1 and the rest equaling to 0. The output matrix $W_{out}$ is constructed by using the linear Moore-Penrose method,

$$W_{out} = YS^\dagger, \tag{5}$$

where † denotes the pseudo-inverse operator. For the evaluation on the testing set of the remaining (10-$N$) utterances, the ten reconstructed outputs corresponding to one digit are averaged over all of the time intervals of $\tau$ of one word, and the digit is identified by taking the maximum value of the ten averaged reconstructed outputs. The recognition rate is obtained by calculating the word success rate. For the recognition rate of each $N$, there is $10!/(N!(10-N)!)$ different ways to pick the $N$ training set; therefore, we average the results from all the different ways to obtain the final recognition rate (cross-validation). The experimental details, the preprocessing and post-processing procedures for the spoken digit recognition task can be found in Torrejon et al.[17].

**Experimental measurements on spintronic oscillator.** The experimental implementation for the spoken digit recognition task is illustrated in Fig. 4a. The preprocessed input signal is generated by a high-frequency arbitrary-waveform generator and injected as a current through the magnetic nano-oscillator. The sampling rate of the source is set to 200 MHz (20 points per interval of time $\theta$). The bias conditions of the oscillator are set by a direct current source ($I_{DC}$) and an electromagnet applying a field ($\mu_0 H$) perpendicular to the plane of the magnetic layers. For the cochlear method, $I_{DC} = 7$ mA, $\mu_0 H = 448$ mT. For the spectrogram method, $I_{DC} = 6$ mA, $\mu_0 H = 430$ mT. See Torrejon et al.[17] for more details.

**Prediction of experimental data.** We use the output signal recorded for every $p = \theta = 100$ ns from the oscillator. The first 50,000 output data points, which corresponds to a time length of 5 ms, from the oscillator of the first utterance of the first speaker and corresponding preprocessed signal as input is used as training set to train a three-layer neural network $f_\theta$ with each hidden layer of 100 units. The trained model is then utilized to predict the response output of the oscillator of all other speakers. The trained $f_\theta$ function is a deterministic function without noise. We therefore evaluate the effect of the noise on the task performance by adding the noise drawn from a Gaussian distribution into the preprocessed input, so that the standard deviation of noise in the output trajectory predicted by Neural ODE ($\sigma_{out}$) is close to the standard deviation of error between experiments and the results of the noiseless trained ODE ($\sigma_{err}$), over the 5 ms training dataset, as shown in Fig. 5b for the cochlear method. See Supplementary Note 5 for the spectrogram method.

**Simulation machine specifications.** For micromagnetic simulations, we used an Nvidia GeForce GTX 1080 graphics processor unit. For Neural ODE simulations, we used an Intel Xeon E5-2640 CPU with 2.5 GHz base clock frequency and 3.0 GHz maximum turbo frequency.

## Data availability

The data that support the findings of this study are available in the github repository [Xing-CHEN18/NeuralODEs_for_physics], at https://github.com/Xing-CHEN18/NeuralODEs_for_physics. The data generated in this study are provided in the Supplementary Information and Source Data file. Source data are provided with this paper.

## Code availability

The micromagnetic simulations are performed using the freely available MuMax3 platform. Neural ODEs simulations are performed using Pytorch. The source codes used in this work are freely available online in the Github repository[76]: https://github.com/Xing-CHEN18/NeuralODEs_for_physics

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

## Acknowledgements

This work was supported by European Research Council Starting Grant NANOINFER (reference: 715872) and BIOSPINSPIRED (reference: 682955). X.C. also acknowledges the support from the China Scholarship Council (No. 201906020155). W.K. also acknowledges the National Natural Science Foundation of China (61871008), Beijing Natural Science Foundation (Grants No. 4202043), and the Beijing Nova Program from Beijing Municipal Science and Technology Commission (Z201100006820042). F.A.A. is a Research Fellow of the F.R.S.-FNRS. The authors would like to thank B. Penkovsky, L. Herrera Diez, A. Laborieux, T. Hirtzlin, and P. Talatchian for discussion and invaluable feedback.

## Author contributions

D.Q. directed the project. X.C. performed all the simulations with the help and advice from D.Q., J.G., F.A.A., M.R., W.K., D.R and W.Z. F.A.A., M.R., J.T., and J.G. provided the experimental measurement data of spintronic oscillators. All authors participated in data analysis, discussed the results and co-edited the manuscript.

## Competing interests

The authors declare no competing interests.
