## [Peer Review File · Nature Communications]

REVIEWER COMMENTS

Reviewer #1 (Remarks to the Author):

The manuscript presents an approach of using Neural Ordinary Differential Equations (NODE) for predicting the dynamic behavior of spintronic structures. The NODE approach used in the manuscript is a modification of the original NODE in two ways: (i) it allows using a reduced set of state variable parameters by using the information from a set of time steps, which replaces the need of using higher-order derivatives and (ii) it allows using time dependent external parameters. The authors show that these modifications allow using the updated NODE approach to the study of several complex problems of using spintronic devices.

The ability to use learning techniques in place of micromagnetic simulations can open various opportunities for analysis and design, which is important for the field of spintronics and nanomagnetism in general.

The presented phenomenology is demonstrated by using it to address several problems, including those based on micromagnetic simulations and those based on experimental data.

The presented methodology is reasonable and the methods are supported by the presented results. One question that could be addressed in more detail is related to the stability. In page 4 of the manuscript, the authors mention that using higher-order derivatives can lead to numerical instabilities and that using multiple time steps instead makes the method stable. The authors should explain why the stability is improved and maybe add a stability proof in the Supplementary Material document. This is because the stability issues may appear in such discretized systems. After all, using information at multiple time steps is numerically equivalent to higher-order derivatives. It is known that higher-order discrete representations for derivatives can be unstable and that ODE systems can be unstable. More attention should be paid to this point.

Another question that one may ask is if the presented method would work for structures in which significantly different physics types may occur in the same system depending on the input parameters. For example, what if the training set only includes switching, whereas a different set of input parameters would lead to sustained oscillations. Micromagnetic simulations would describe such a behavior. Would it be possible to catch with NODE? Can the authors address this point. That is, it would be important to add a discussion on limitations of the presented approach.

There is a review of existing literature. However, some recent works related to using neural networks and machine learning in computational magnetism are missing. In particular, the authors are advised to review the works by the groups of Schrefl and Sues. Here are two examples:

1) Markus Gusenbauer, Harald Oezelt, Johann Fischbacher, Alexander Kovacs, Panpan Zhao, Thomas George Woodcock & Thomas Schrefl, "Extracting local nucleation fields in permanent magnets using machine learning," npj Computational Materials volume 6, Article number: 89 (2020).

2) Alexander Kovacs, Johann Fischbacher, Harald Oezelt, Markus, Gusenbauer, Lukas Exl, Florian Bruckner, Dieter Suess, Thomas Schrefl, "Learning magnetization dynamics,"

Journal of Magnetism and Magnetic Materials, Volume 491, 1 December 2019, 165548.

Reviewer #2 (Remarks to the Author):

In this manuscript, the authors present a novel method for predicting the behavior of spintronic devices based on neural networks. The authors modified the neural ODE to be applied to the simulation of the spintronic system and then performed several simulations to evaluate their novel method. The result of simulations using the authors' novel method well agreed with the results based on conventional micromagnetic simulation, indicating the novel method can be used to simulate the dynamics of spintronic devices. In particular, the calculation time of this novel method is much faster than that of the conventional micromagnetic simulation. The long calculation time of micromagnetic simulation is a long-standing issue. Hence, the authors' novel method can be widely used to simulate the long-time dynamics of spintronic devices. Therefore, although there are a few minor unclear points for me as described below, I think it is worth publishing in Nature Communications.

1 In the simulation in single skyrmion dynamics [Fig. 2(c)], the authors claim that the models with $k \geq 2$ are trained well because the system can be described by two variables. At a glance, it makes sense. However, in the simulation in multi skyrmion systems [Fig. 2(d)], the model with $k = 2$ also is well trained, although more than two variables are perhaps required to describe the multi skyrmion system. Why is the model with $k = 2$ trained well even in the case of the multi-skyrmion system?

2 In Fig1, the authors demonstrate the result of the simulation of the skyrmion system, in which the predicted training output well agree with the ideal output. In my understanding, all data from 0 to 50 ns were used in training. Can this model predict the data that was not used for training well? I am curious about overlearning although it probably does not matter since the simulation in Fig3 works well.

3 The authors demonstrate the simulation of skyrmions and spin-torque nanoscillators. Can this method simulate the dynamics of other systems such as ferromagnetic domains? I would like to

know whether there is any limitation nor not. For example, how much does the system size affect the simulation time or prediction accuracy?

Reviewer #3 (Remarks to the Author):

Overall, the authors have proposed a mathematically sound extension of neural ordinary differential equations (NODE, Chen, et al, 2018), but the manuscript requires some overhauling to make it clear what authors novel contributions are. Many important references on neural ODEs are missing.

Here are specific comments:

Results Section:

1) variational formulation including higher-order derivatives:

- Lifting NODEs to higher orders has been studied in Dissecting Neural ODEs (Massaroli, et al, 2020).
- As this formulation has not been used in any of experiments, but has been only used to motivate why time-delayed embedding is used, the paragraphs explain the mathematical formulations could go to supplementary materials.

2) Time-delay embedding (TDE):

- using TDE's time-series modeling is a well-known concept and it is hard to say that it is the main contribution of the manuscript. The authors should give more credit to the original paper in the main manuscript, not just citing the original paper in the supplemental material:

'Sugihara, G. and May, R. M. Nonlinear forecasting as a way of distinguishing chaos from measurement error in time series. *Nature*, 344(6268):734–741, 1990. ISSN 1476-4687'.

- Also, $y_1(t)$, $t_1(t+\Delta t)$, ... : using positive Δt is confusing as $y(0) = (y_1(0), y_1(\Delta t), y_1(2\Delta t) \dots)$ does not seem to use delayed signals. Normally, time delay embedding is written as $y(0) = y_t, y_{t-1}, \dots$.

3) NODE extension to take external inputs

An extension of NODEs to include external inputs has been studied in the literature.

- Increasing the dimension of state space by augmenting extra state variables:

Augmented Neural ODEs, Dupong et al, NeurIPS, 2019

- Adding external inputs to NODEs in ECG dynamics modeling

ECG ODE-GAN: Learning Ordinary Differential Equations of ECG Dynamics via Generative Adversarial Learning, Golany, et al, AAAI 2021.

- Adding external inputs to NODEs in computational physics problems

Parameterized Neural Ordinary Differential Equations: Applications to Computational Physics Problems, Lee and Parish, 2021

=====

On robustness of the proposed method on noisy inputs:

I am not convinced by the authors' claim that the proposed method is for noisy inputs. More explanation on why this is the case would be needed.

=====

Questions on experiments:

Training

- not clear how is the data composed? Is this a single trajectory? how was it generated? how many samples in it? If multiple trajectories are considered, how did they generated by varying initial conditions? varying the parameters? How is the train/val/test composed?

- Had the authors compared their model against the original neural ODEs?

=====

(Clarification) Algorithm 1: it seems that the authors just train the neural ODEs with training data by using Algorithm 1 and then use the trained model to perform the prediction task. So there is no training algorithm designed specifically for prediction in future.

=====

(Minor) Confusing terms

Neural ODE, NeurODE, SPIN-NEURAL ODE: are they indicating the same thing?

In conclusion, I believe the authors should make it clear that the novelty of the manuscript is in that the state-of-the-art deep-learning technique for time-series modeling has been applied to complex real-world physical processes, but not designing a novel extension of NODEs.

Response to Reviewers

Reviewer #1

The manuscript presents an approach of using Neural Ordinary Differential Equations (NODE) for predicting the dynamic behavior of spintronic structures. The NODE approach used in the manuscript is a modification of the original NODE in two ways: (i) it allows using a reduced set of state variable parameters by using the information from a set of time steps, which replaces the need of using higher-order derivatives and (ii) it allows using time dependent external parameters. The authors show that these modifications allow using the updated NODE approach to the study of several complex problems of using spintronic devices.

The ability to use learning techniques in place of micromagnetic simulations can open various opportunities for analysis and design, which is important for the field of spintronics and nanomagnetism in general.

The presented phenomenology is demonstrated by using it to address several problems, including those based on micromagnetic simulations and those based on experimental data.

We would like to thank the reviewer for his/her time and comments, which have allowed us to improve the quality of our manuscript. We acknowledge the reviewer for recognizing our work as important for opening various opportunities for analysis and design in the field of spintronics and nanomagnetism in general. We have addressed the points raised by the reviewer and revised the manuscript accordingly.

Details

Q1 The presented methodology is reasonable and the methods are supported by the presented results. One question that could be addressed in more detail is related to the stability. In page 4 of the manuscript, the authors mention that using higher-order derivatives can lead to numerical instabilities and that using multiple time steps instead makes the method stable. The authors should explain why the stability is improved and maybe add a stability proof in the Supplementary Material document. This is because the stability issues may appear in such discretized systems. After all, using information at multiple time steps is numerically equivalent to higher-order derivatives. It is known that higher-order discrete representations for derivatives can be unstable and that ODE systems can be unstable. More attention should be paid to this point.

As the reviewer mentioned, the method of using the information at multiple time steps and the method of using higher-order representations for training Neural ODE appear essentially equivalent. However, they do not work equally well in practice. To clarify this point, we have included more discussion within the main paper. We have also compared explicitly the two methods on a demonstrative example in Supplementary Note 3. We summarize this new material here.

The major issue is that for a measured noisy time series, the noise amplitude increases by taking derivatives. The detailed proof regarding the noise to signal ratio can be found in section B of Supplementary Note 8.

To illustrate the difficulty of using computed derivatives to train Neural ODEs, we add noise to the training set generated from the micromagnetic simulations of the one-skyrmion system. We first inject a noise $n(t)$ with zero mean and standard deviation $\sigma_{noi} = 0.5$ into the clean training set $y(t)$ to have the noisy observation $s(t) = y(t) + n(t)$. The variance of the signal $y(t)$ extracted from simulations is $\sigma_{sig}^2 = E[y(t)^2] - E[y(t)]^2 = 1.22$. Therefore, the noise-to-signal ratio is $\sigma_{noi}/\sigma_{sig} = 0.45$.

Using the same method, and taking for the derivative of the signal $\dot{y}(t) = (y(t + \Delta t) - y(t - \Delta t))/(2\Delta t)$ and for the derivative of the noisy signal as $\dot{s}(t) = (s(t + \Delta t) - s(t - \Delta t))/(2\Delta t)$, we obtain from simulations that the noise to signal ratio of the first-order derivative $\dot{s}(t)$ is $\sigma_{noi,der}/\sigma_{sig,der} = 3.22$, which is much larger than that of the original signal.

We now compare the training performance of the time-derivative and time-delayed based methods. We use the information at the three continuous time points $(t, t - \Delta t, t + \Delta t)$. For the time-delay method, we consider a vector of variables $\mathbf{s}(t) = (s(t - \Delta t), s(t), s(t + \Delta t))$. For the time-derivative method, the vector $\mathbf{s}(t) = (s(t), (s(t + \Delta t) - s(t - \Delta t))/(2\Delta t))$ is used. Figs. S4a and b show the training loss (mean square error, MSE) as a function of iterations for $\sigma_{noi} = 0$ (without noise) and $\sigma_{noi} = 0.5$, respectively. It is observed that both methods show similar training performance w.r.t accuracy when noise is absent from the system, however, the time derivative method performs worse than the time delay method when noise exists. This is reasonable because the noise is amplified during the calculation of the first-order derivative, and thus the convergence becomes difficult.

We have now commented on this issue in the revised version of the main text. We have also added the new Supplementary Note 3 and the associated new Supplementary Figure 4 (also reproduced here) to discuss and compare the two methods and the impact of noise on their performance.

Supplementary Figure 4. Comparison between the time-delay and time-derivative methods in terms of training performance. Training loss (mean square error, MSE) as a function of iterations for **a** $\sigma_{noi} = 0$ (without noise) and **b** $\sigma_{noi} = 0.5$, for the one-skyrmion system.

The added paragraph in the main text reads:

'The use of delayed variables may seem equivalent to the use of time derivatives, as the latter are typically calculated by taking linear combinations of discrete samples of the data. However, as mentioned earlier, numerical derivatives amplify the noise present in the training data, making the Neural ODE training process much more difficult with time derivatives than with delayed variables. Supplementary note 3 discusses this issue in detail and provides an example comparing these two training techniques.'

Q2 Another question that one may ask is if the presented method would work for structures in which significantly different physics types may occur in the same system depending on the input parameters. For example, what if the training set only includes switching, whereas a different set of input parameters would lead to sustained oscillations. Micromagnetic simulations would describe such a behavior. Would it be possible to catch with NODE? Can the authors address this point. That is, it would be important to add a discussion on limitations of the presented approach.

We thank the reviewer for pointing out this question. It is possible to train a Neural ODE modeling a structure exhibiting abrupt switching or oscillations, depending on the value of input parameters, if both situations are included within the training set. To illustrate this capability, we have now simulated a situation where a spin valve may switch or generate sustained oscillations depending on the value of the applied external magnetic field. We have included this material as a new Supplementary Note 6. We reproduce this Note here.

The training data is obtained from micromagnetic simulations of a spin-valve nanopillar, under the influence of a spin current polarized out-of-plane, and an out-of-plane magnetic field $H(t)$ applied in the z direction. The magnetization of the pinned layer is fixed along the negative z axis. The diameter and the thickness of the nanopillar are 80 nm and 2 nm, respectively. The key parameters used in the simulations are: saturation magnetization $M_s = 1.2$ MA/m, exchange stiffness $A = 20$ pJ/m, interfacial perpendicular magnetic anisotropy $K_u = 0.5$ MJ/m³, and damping constant $\alpha = 0.1$. In the simulations, a constant electric direct current is applied with a current density of 10^{11} A/m². In the training dataset, artificial sinusoidal waveform of magnetic field with a variety of random amplitudes are applied sequentially as input, as shown below in Fig. S8a. In these artificial training conditions, the structures exhibits relatively complex and difficult-to-interpret behaviors, but which allow training a Neural ODE that is valid in all regimes. For this purpose, we use a three-dimensional vector containing the information (m_x, m_y, m_z) . A total length of 500-nanoseconds of dynamics is used to train the system. The trained results of m_z (dashed orange curve) and m_x (dashed olive curve), shown in Fig. S8a, demonstrate excellent agreement with the micromagnetic simulations.

In the test dataset, we apply a time-varying magnetic field at three different initial conditions of magnetization, covering a wide range of situations where the system is expected to exhibit either switching or sustained oscillations.

- *Firstly, the magnetization is aligned in the positive z direction by applying an external field of $\mu_0 H = 0.65$ T initially. A subsequent decreasing field is applied till a constant value to switch the magnetization (e.g., $\mu_0 H = -0.5$ T) or to generate a sustained oscillation (e.g., $\mu_0 H = -0.1, 0.2, 0.3$ T), as shown in the left column of Fig. S9a for different constant values.*
- *Secondly, the magnetization is aligned in the negative z direction by applying an external field of $\mu_0 H = -0.5$ T initially, a subsequent decreasing field is applied till a constant value to switch the magnetization (e.g., $\mu_0 H = 0.7$ T) or to generate a sustained oscillation (e.g., $\mu_0 H = 0.5, 0.2, -0.2$ T), as shown in the middle column of Fig. S9a for different constant values.*
- *In the third scenario, no external field and only the constant electric current is applied initially. In this situation, the system exhibits self-sustained magnetization oscillations. A subsequent decreasing or increasing field is applied till a constant value to switch the magnetization (e.g., $\mu_0 H = -0.5, 0.7$ T), as shown in the right column of Fig. S9a for different constant values.*

We can see that in this test dataset, the results of Neural ODE and micromagnetic simulations are again in excellent agreement. The Neural ODE is able to predict all the various behaviors, equivalently to micromagnetic simulations.

For clearer comparisons, we further extract the oscillation frequency at different magnetic fields based on the results from Neural ODE and the micromagnetic simulations, as shown in Fig. S9b. This example demonstrates that the Neural ODE is able to describe different dynamical behaviors (switching and sustained oscillations) under different input parameters.

It should be stressed that this structure features a deterministic dynamical process. One of the important limitations of using Neural ODE to model the micromagnetic simulation data is that Neural ODE should be trained with data set from such a deterministic dynamical process. If a structure features a profoundly stochastic dynamical process, as is sometimes observed in room-temperature

switching of magnetic tunnel junction or domain wall motion in some regimes, it would be difficult for the current Neural ODE model to capture the law behind the training set. For clearer explanations, we have added a discussion of this limitation of Neural ODEs in the revised version of the main text:

'Last but not least, Neural ODEs can be used for modeling systems featuring different types of behaviors, provided that examples of the different behaviors were included in the training dataset. Supplementary Note 6 shows an example, where a single Neural ODE can model a device, which depending on the value of the external magnetic field, exhibits either a switching or a sustained oscillatory behavior. A limitation of Neural ODE, however, is that they cannot be trained to model systems exhibiting profoundly stochastic behavior, as is sometimes observed in room-temperature switching of magnetic tunnel junction [66] or domain wall motion in some regimes [67]. Neural ODEs are adapted for systems obeying deterministic equations. Future work regarding the modeling of stochastic behaviors of a physical system using Neural ODE remains to be explored, which could rely on recent developments of stochastic Neural ODE theory [68,69].'

Supplementary Figure 8. Abrupt switching and oscillations of a spin-valve nanopillar modeled by Neural ODE (train dataset). **a** Input and output dynamics in the training dataset. The green curve shows the random sinusoidal input waveform of magnetic field with a variety of amplitudes applied in the z direction. The blue and red curves show the mean output magnetization of x component (m_x) and z component (m_z), respectively, obtained from micromagnetic simulations. The dashed orange (m_x) and dashed olive curves (m_z) are the corresponding trained results of Neural ODE.

Supplementary Figure 9. Abrupt switching and oscillations of a spin-valve nanopillar modeled by Neural ODE (test dataset). **a** Neural ODE test results under different input parameters of the external magnetic field and at three different initial conditions of magnetization (from left to right). In the left column, the magnetization is aligned in the positive z direction by applying an external field of $\mu_0H = 0.65$ T initially, a subsequent decreasing field is applied till a constant value to switch the magnetization ($\mu_0H = -0.5$ T) or to generate a sustained oscillation ($\mu_0H = -0.1, 0.2, 0.3$ T). In the middle column, the magnetization is aligned in the negative z direction by applying an external field of $\mu_0H = -0.5$ T initially, a subsequent increasing field is applied till a constant value to switch the magnetization ($\mu_0H = 0.7$ T) or to generate a sustained oscillation ($\mu_0H = 0.5, 0.2, -0.2$ T). In the right column, no external field and only the constant electric current is applied initially, the system exhibits self-sustained magnetization oscillations. A subsequent decreasing or increasing field is applied till a constant value to switch the magnetization ($\mu_0H = -0.5, 0.7$ T). The blue and red curves show the mean output magnetization of x component (m_x) and z component (m_z), respectively, from micromagnetic simulations. The dashed orange (m_x) and dashed olive curves (m_z) are the corresponding trained results of Neural ODE. **b** Oscillation frequency F at different magnetic fields obtained by using the results from Neural ODE (orange star) and the micromagnetic simulations (blue curve). The red cross symbols represent the switching behavior at the specific amplitude of magnetic fields obtained by both Neural ODE and the micromagnetic simulations.

Q3 There is a review of existing literature. However, some recent works related to using neural networks and machine learning in computational magnetism are missing. In particular, the authors are advised to review the works by the groups of Schrefl and Suess. Here are two examples:

1) Markus Gusenbauer, Harald Oezelt, Johann Fischbacher, Alexander Kovacs, Panpan Zhao, Thomas George Woodcock & Thomas Schrefl, "Extracting local nucleation fields in permanent magnets using machine learning," npj Computational Materials volume 6, Article number: 89 (2020).

2) Alexander Kovacs, Johann Fischbacher, Harald Oezelt, Markus, Gusenbauer, Lukas Exl, Florian Bruckner, Dieter Suess, Thomas Schref, "Learning magnetization dynamics," Journal of Magnetism and Magnetic Materials, Volume 491, 1 December 2019, 165548.

We have reviewed the relevant recent works about machine learning in computational magnetism and added them in the introduction of the revised manuscript:

'In the field of nanomagnetism and micromagnetics, deep neural networks are used to extract microstructural features [31–34] in magnetic thin film elements, and to explore materials with ease [35]. Refs.[36–38] use a sophisticated combination of machine learning techniques to predict the magnetization dynamics of magnetic thin film elements over one nanosecond.'

Reviewer #2

In this manuscript, the authors present a novel method for predicting the behavior of spintronic devices based on neural networks. The authors modified the neural ODE to be applied to the simulation of the spintronic system and then performed several simulations to evaluate their novel method. The result of simulations using the authors' novel method well agreed with the results based on conventional micromagnetic simulation, indicating the novel method can be used to simulate the dynamics of spintronic devices. In particular, the calculation time of this novel method is much faster than that of the conventional micromagnetic simulation. The long calculation time of micromagnetic simulation is a long-standing issue. Hence, the authors' novel method can be widely used to simulate the long-time dynamics of spintronic devices. Therefore, although there are a few minor unclear points for me as described below, I think it is worth publishing in Nature Communications.

We would like to thank the reviewer for his/her time and comments, which have allowed us to improve the quality of our manuscript. We thank the reviewer for his/her review and his/her appreciation of our work as worth publishing in Nature Communications. We have addressed the points raised by the reviewer and revised the manuscript accordingly.

Q1. 1 In the simulation in single skyrmion dynamics [Fig. 2(c)], the authors claim that the models with $k \geq 2$ are trained well because the system can be described by two variables. At a glance, it makes sense. However, in the simulation in multi skyrmion systems [Fig. 2(d)], the model with $k = 2$ also is well trained, although more than two variables are perhaps required to describe the multi skyrmion system. Why is the model with $k = 2$ trained well even in the case of the multi-skyrmion system?

In both cases of one-skyrmion and multi-skyrmions system, the skyrmion dynamics, i.e., the oscillations of skyrmion sizes under external excitations is related to the breathing mode, which can be modeled by two variables [2]. For a multi-skyrmions system, the skyrmions show coherent oscillations, i.e., all skyrmions oscillate in phase with the same frequency [1]. Grain inhomogeneity mainly distorts the shapes of skyrmions. Therefore, the averaged magnetization dynamics Δm_z can be described similarly to the idealized one-skyrmion system [2].

For clearer explanation, we have added the following statement in the revised version of the main text:

'In this case, the skyrmions show coherent oscillations, i.e., all skyrmions oscillate in phase with the same frequency [49], and grain inhomogeneity mainly distorts the shapes of skyrmions. Therefore, the averaged magnetization dynamics can be described in the same way as the idealized one-skyrmion system, for any dimension $k \geq 2$.'

[1]. Tejo, Felipe, et al. "Oscillations of skyrmion clusters in Co/Pt multilayer nanodots." Scientific reports 10.1 (2020): 1-8.

[2]. Chen, Xing, et al. "Magnetic skyrmion spectrum under voltage excitation and its linear modulation." Physical Review Applied 12.2 (2019): 024008.

Q2. In Fig1, the authors demonstrate the result of the simulation of the skyrmion system, in which the predicted training output well agree with the ideal output. In my understanding, all data from 0 to 50 ns were used in training. Can this model predict the data that was not used for training well? I am

curious about overlearning although it probably does not matter since the simulation in Fig3 works well.

Yes, the input and output dynamics from 0-50 ns shown in Fig. 1d-e are all used for the training. For the test results shown in Fig. 1f-g, however, we used very different waveforms of inputs, which are not included in the training set. Test inputs are composed of a pulse signal of ΔK_u (ΔD) and a constant value of D (K_u) to obtain the response magnetization dynamics Δm_z and thus to predict the response frequency for a specific material parameter D and ΔK_u , and are thus different from the sinusoidal waveforms of the time-varying input parameters of ΔD and ΔK_u shown in Fig. 1d.

Additionally, the test results shown in Fig. 3 are indeed another example where the test dataset differs fundamentally from the training dataset. In the test dataset, the amplitude of the time-varying input voltage (ΔK_u) is the pre-processed signal of the Mackey-Glass time series, while the input voltage in the train set is a random sine voltage (ΔK_u) with a variety of amplitudes. In these various testing situations, the output trajectory Δm_z simulated by micromagnetic simulations and predicted by Neural ODE are in very good agreement.

Therefore, the trained Neural ODE model is able to predict data that was not used for training. This comes from the fact that the diversity of inputs and output dynamics in the train set, even if it is only 50-ns long, was enough for the Neural ODE to capture the physical model during the training without overfitting.

These points are now more clearly stated in the paper, which reads,

'In that case, the test inputs are composed of a pulse signal of ΔK_u (or ΔD) and a constant value of D (or K_u) to induce an oscillating magnetic response Δm_z and thus to predict the corresponding frequency for specific material parameters D and K_u . They are thus different from the sinusoidal waveforms of the ΔD and ΔK_u used for training (Fig. 1d), which is important to test the ability of the neural network to generalize'

Q3. The authors demonstrate the simulation of skyrmions and spin-torque nanoscillators. Can this method simulate the dynamics of other systems such as ferromagnetic domains? I would like to know whether there is any limitation nor not. For example, how much does the system size affect the simulation time or prediction accuracy?

Yes, this method can be used to simulate a variety of systems. In our response to Q2 of Reviewer #1, e.g., we trained a Neural ODE to model a structure, which depending on the magnetic field, exhibits a switching or sustained oscillatory behavior.

In terms of limitations, one of the important limitations of using Neural ODE to model the micromagnetic simulation data is that Neural ODE should be trained with data set from deterministic process of dynamics; that is, if the training set is taken from a profoundly stochastic dynamical process, it would be difficult for the current Neural ODE model to capture the law behind the training set. This can be a challenge to model the dynamics of some domain wall structures at room temperature, where domain wall motion is intrinsically a stochastic process [1].

[1] Meier, Guido, et al. "Direct imaging of stochastic domain-wall motion driven by nanosecond current pulses." *Physical review letters* 98.18 (2007): 187202.

Concerning the simulation time and accuracy:

System size is an important factor affecting simulation time in micromagnetic simulations. For a large system, it might be a challenge to obtain enough training data to train a Neural ODE. However, in the

trained Neural ODE itself, system size is not a factor influencing the simulation time as long as the system dynamics can be described by a set of state variables. The simulation time is determined by the number of data points to be predicted: the prediction accuracy is determined by how good is the model being trained and system size is not directly an influencing factor.

We have added the following clarification in the main article:

'Last but not least, Neural ODEs can be used for modeling systems featuring different types of behaviors, provided that examples of the different behaviors were included in the training dataset. Supplementary Note 6 shows an example, where a single Neural ODE can model a device which, depending on the value of the external magnetic field, exhibits either a switching or a sustained oscillatory behavior. A limitation of Neural ODE, however, is that they cannot be trained to model systems exhibiting profoundly stochastic behavior, as is sometimes observed in room-temperature switching of magnetic tunnel junction [66] or domain wall motion in some regimes [67]. Neural ODEs are adapted for systems obeying deterministic equations. Future work regarding the modeling of stochastic behaviors of a physical system using Neural ODE remains to be explored, which could rely on recent developments of stochastic Neural ODE theory [68,69]'

Concerning speed, we have added this clarification in the Discussion section:

"The Neural ODE simulation time does not dependent directly on the size of the system, but only on the number of dimensions and the number of data points to be predicted; the possibility to train a Neural ODE is therefore mostly determined by the availability of appropriate training data."

Reviewer #3

Overall, the authors have proposed a mathematically sound extension of neural ordinary differential equations (NODE, Chen, et al, 2018), but the manuscript requires some overhauling to make it clear what authors novel contributions are. Many important references on neural ODEs are missing.

We would like to thank the reviewer for his/her time and comments, which have allowed us to improve the quality of our manuscript. We have overhauled the manuscript to clarify the novel contributions of this work, and have also added the missing references on Neural ODEs. We have addressed the points raised by the reviewer and revised the manuscript accordingly.

Q1 Here are specific comments:

Results Section:

1) variational formulation including higher-order derivatives:

- Lifting NODEs to higher orders has been studied in Dissecting Neural ODEs (Massaroli, et al, 2020).
- As this formulation has not been used in any of experiments, but has been only used to motivate why time-delayed embedding is used, the paragraphs explain the mathematical formulations could go to supplementary materials.

We have moved the paragraphs explaining the mathematical formulations of the higher-order ODE to Supplementary Note 7 as suggested. We have also added the reference (Massaroli, et al, 2020) into the Discussion section, as part of:

'extensions of Neural ODEs at the second-order [60] or higher-order [61], can learn the low-dimensional physical dynamics of the original system. .'

Q2 2) Time-delay embedding (TDE):

- using TDE's time-series modeling is a well-known concept and it is hard to say that it is the main contribution of the manuscript. The authors should give more credit to the original paper in the main manuscript, not just citing the original paper in the supplemental material:

'Sugihara, G. and May, R. M. Nonlinear forecasting as a way of distinguishing chaos from measurement error in time series. *Nature*, 344(6268):734–741, 1990. ISSN 1476-4687'.

As part of several changes to clarify the contribution of the paper, we have given more credit to the prior work about the delay embedding theorem in the introduction section, results section and discussion section, and have also cited the original paper in the main paper. The paragraph in the introduction section reads:

'In the rest of the paper, we first explain how we modified Neural ODE in order to be able to train the whole set of parameters based on the temporal evolution of a single physical variable of the nanodevice under the effect of fluctuating inputs. For this purpose, we have integrated in the Neural ODE framework the idea of the embedding theorem for the reconstruction of the state space from a time series.'

The paragraph in the Results section reads:

'More specifically, the theorems by Takens [43] and by Sauer et al. [44] state that if the sequence $\mathbf{y}(t)$ consists of scalar measurements of the state vector of a dynamical system, then under certain genericity assumptions, the time delay embedding provides a one-to-one image of the original set, provided k is large enough. The prevalent application of employing delay embedding is to make short term predictions of nonlinear time series [45, 46]. The combination of Neural ODE with the delay embedding theorem enables making predictions for nonlinear time series with arbitrary lengths in a precise way, because the neural network provides a strong language to describe the system non-linearity and thus the physical pattern can be captured through training with a large number of observed data.'

The paragraph in the discussion section reads:

'We emphasize here that our idea is closely related to the classical theorem of time delay embedding for state space reconstruction, where the past and future of a time series containing the information about unobserved state variables can be used to define a state at the present time. The theorem was widely applied for forecasting in many real world engineering problems [45, 46, 65], but was largely restricted into making very short term predictions for lack of modeling frame specifically designed for time series. The advent of Neural ODEs, whose formalism naturally incorporates time series, allows predictions of arbitrary length and high accuracy to be made by training a system equivalent to the original physical system.'

- Also, $y_1(t), t_1(t+\Delta t), \dots$: using positive Δt is confusing as $y(0) = (y_1(0), y_1(\Delta t), y_1(2\Delta t) \dots)$ does not seem to use delayed signals. Normally, time delay embedding is written as $y(0) = y_t, y_{t-1}, \dots$.

We chose this notation, as the idea of using negative Δt in TDE is for predicting the state value at a future time by using the past information. In our work, we use a vector containing multiple states at successive time points as an equivalent representation of the dynamics of the original state variables for training an equivalent system, and using a positive Δt felt more natural. There are other relevant works using delay vector with positive Δt for state space reconstruction [1-6]. However, our scheme works with negative Δt equivalently.

[1] Abarbanel, Henry DI, and Matthew B. Kennel. "Local false nearest neighbors and dynamical dimensions from observed chaotic data." *Physical Review E* 47.5 (1993): 3057.

[2] Abarbanel, Henry DI, et al. "The analysis of observed chaotic data in physical systems." *Reviews of modern physics* 65.4 (1993): 1331.

[3] Cao, Liangyue. "Practical method for determining the minimum embedding dimension of a scalar time series." *Physica D: Nonlinear Phenomena* 110.1-2 (1997): 43-50.

[4] Kim, H_S, R. Eykholt, and J. D. Salas. "Nonlinear dynamics, delay times, and embedding windows." *Physica D: Nonlinear Phenomena* 127.1-2 (1999): 48-60.

[5] Gao, J. B. "Recurrence time statistics for chaotic systems and their applications." *Physical Review Letters* 83.16 (1999): 3178.

[6] Tolle, Charles R., and Mark Pengitore. "Phase-space reconstruction: a path towards the next generation of nonlinear differential equation based models and its implications towards non-uniform sampling theory." 2009 2nd International Symposium on Resilient Control Systems. IEEE, 2009.

For clarification, we have now added this footnote in the main text:

'We chose a positive Δt_d value in our work. Using a negative Δt_d value, as is usually done in the time-delay embedding literature, leads to equivalent results.'

Q3 3) NODE extension to take external inputs

An extension of NODEs to include external inputs has been studied in the literature.

- Increasing the dimension of state space by augmenting extra state variables:

Augmented Neural ODEs, Dupong et al, NeurIPS, 2019

- Adding external inputs to NODEs in ECG dynamics modeling

ECG ODE-GAN: Learning Ordinary Differential Equations of ECG Dynamics via Generative Adversarial Learning, Golany, et al, AAAI 2021.

- Adding external inputs to NODEs in computational physics problems

Parameterized Neural Ordinary Differential Equations: Applications to Computational Physics Problems, Lee and Parish, 2021.

We have added these references into the discussion section of the main text, which now reads,

'Neural ODE integrated with external inputs has also been studied in some previous literature [62–64]. Augmented Neural ODEs [63] solve the initial value problem in a higher-dimensional space, by concatenating each data point with a vector of zeros to lift points into the additional dimensions. This strategy avoids trajectories intersecting each other, and thus allows modeling more complex functions using simpler flows, while achieving lower losses, reducing computational cost, and improving stability, and generalization. Parameterized Neural ODE [64] extends Neural ODEs to have a set of input parameters that specify the dynamics of the Neural ODEs model such that the dynamics of each trajectory are characterized by the input parameter instance.'

Q4 On robustness of the proposed method on noisy inputs:

I am not convinced by the authors' claim that the proposed method is for noisy inputs. More explanation on why this is the case would be needed.

We have now added more demonstrations about the training performance when different levels of noise are injected into the system, in Supplementary Note 2 and the associated new Supplementary Figure 3, (also shown below). We also now discuss with more details in the paper why the method applies to noisy inputs into the Neural ODE

- More explanations about why the method applies to noisy inputs into the Neural ODE:

'For the modeling of noisy time series, a dimension of two is insufficient to train a good model, and higher accuracy can be obtained by increasing the number of delays. This result can be explained by the fact that gathering more information, i.e., adopting a delay vector of higher dimension means less distortion and lower noise amplification when the time delay embedding is mapped into the original state space [48]'

- More explanations and demonstrations about the training performance when different levels of noise are added into the system in Suppl. Note 2 and associated new Suppl. Fig. 3:

'This note investigates the training of Neural ODEs over noisy data. The strategy for treating noisy dynamical systems comes from the idea, present in the time-delay embedding for state space

reconstruction, that increasing the dimension of the reconstructed space can typically decrease the distortion of noise distribution and reduce the amplification of noise level when the delay embedding is transformed to the original state space¹. Normally, the prediction errors depend on both the noise amplification and the estimation error, which depends on the method of approximation, i.e., the model itself. For most good approximation schemes, the estimation error can be close to zero in the limit of a large number of data points. The prediction errors in this limit are entirely due to the noise. Therefore, by increasing the dimension of the delay vector, a more precise model can be learned.

We use the simulation data of the one-skyrmion system as a demonstration. We obtain noisy training data by artificially adding random Gaussian noise with mean $\mu=0$ and standard deviation $\sigma_{noi} = 0.1, 0.5$ and 1 into the normalized training data (Δm_z), as shown in Fig. S3a. Fig. S3c presents the training loss (MSE) as a function of iterations, obtained when training Neural ODEs of dimension $k=2,4,8,12$ over these noisy data with different standard deviations. These results differ from the training results obtained in the noiseless system (see Fig. S2c), where $k \geq 2$ guarantees a sufficiently good model to be trained. Here, the larger the k , the better the model is. On the other hand, training Neural ODE with a large k value (e.g., for $k=12$) does not always ensure stable convergence, as seen in the low noise system, because the delayed system dynamics becomes a stiff problem to be solved in the high dimensional space, therefore, it is necessary to choose an appropriate k for training. Further, we remark that the fitted standard deviations of the training error distribution σ_{err} of the trained Neural ODEs, compared to the true trajectories (without noise), gradually approach the standard deviation of the added noise σ_{noi} when k is increased (see Fig. S3b for $\sigma_{noi}=0.1$). Fig. S3d shows the testing results of the Neural ODE for $k=2$ and $k=12$ under different standard deviations of the added noise. Here, the Mean Square Error (MSE) showed in the title of each graph in Fig. S3d is the MSE between the testing trajectory and the true trajectory without noise. The MSE approaches zero, which justifies a good training system, as k increases especially for the low noise system (e.g., for $\sigma_{noi}=0.1$). However, as the noise level grows, increasing the k value still helps in making more precise prediction with smaller MSE value, it becomes more difficult to lower the error to zero compared to the true trajectory even with a relatively large k value. A total number of data points $n=15,000$ are used for training and testing, respectively.

Still, the effect of noise is very complicated in many real world applications, the demonstration presented here provides a perspective of the possibility of making better prediction when only limited information of the noise is known. There are still many other ways to help distinguish the real system dynamics from the dynamics contaminated with noise, for example, by conjoining different filters with the reconstruction scheme.'

Supplementary Figure 3. Training performance of Neural ODE model using incomplete noisy data. **a** Schematic graph of the procedure to obtain training time series with noise, by artificially adding random values drawn from a Gaussian distribution with mean $\mu=0$ and standard variance $\sigma_{noi} = 0.1, 0.5$ and 1 respectively into the output time series of the normalized Δm_z . **b** Training error distribution of the trained Neural ODE compared to the true trajectory without noise for different k values where $\sigma_{noi}=0.1$. **c** Training loss (MSE) as a function of iterations for $\sigma_{noi} = 0.1, 0.5$ and 1 , respectively. **d** Final predicted testing results of Neural ODE for $k=2$ and $k=12$ under different standard deviation of the added noise. The Mean Square Error (MSE) showed in the title of each graph is the MSE between the testing trajectory and the true trajectory without noise.

Q5 Questions on experiments:

Training

- not clear how is the data composed? Is this a single trajectory? how was it generated? how many samples in it? If multiple trajectories are considered, how did they generated by varying initial conditions? varying the parameters? How is the train/val/test composed?
- Had the authors compared their model against the original neural ODEs?.

- About the data composition.

The training data is a single trajectory generated by continuously varying the input parameters.

Specifically, for the results presented in Fig. 1, the training dataset is generated by applying a sequence of sinusoidal waveforms of ΔD and ΔK_u with random amplitudes (as shown in Fig. 1a) as inputs into the skyrmion system to obtain the corresponding trajectory of dynamics Δm_z (as shown in Fig. 1e). A total number of 15,000 data points are generated for training. During the training, the first $n = 10,000$ is

used for training and the last 5,000 data points is used for validation. The test set trajectories are generated by applying a pulse signal of ΔK_u (or ΔD) and a constant value of ΔD (or ΔK_u) to obtain the corresponding trajectory of dynamics Δm_z for further calculation of response frequency (shown in Fig. 1f-g). A total number of 37 test trajectories are generated for different initial values of D and K_u , and each test trajectory is composed of 800 data points.

For the results presented in Fig. 2, only ΔK_u is applied as varying input in the form of sinusoidal wave with random amplitudes into the skyrmion system, and the corresponding output trajectory Δm_z is recorded. A total number of data points 15,000 and 20,000 are generated for training one skyrmion system and multi-skyrmions system respectively. During the training, the first $n = 10,000$ and $n = 15,000$ are used for training, and the last 5,000 data points is used for validation. The test set trajectory is generated by applying a time varying ΔK_u in the form of preprocessed Mackey-Glass time series (as shown in Fig. S5a-b in the Suppl. Note 3).

Fig. 3 is using the model trained in Fig. 2. Therefore, the Mackey-Glass time series, composed of 500,000 data points, are all used for test.

The Neural ODEs trained on experimental data (Fig. 4) use 50,000 data points for training, and 10,000 for validation. The training set has 522,8800 points in the cochlear case and 850,7200 points in the spectrogram case.

- About the original form of Neural ODE.

The original Neural ODE fails to learn if we consider the target trajectory as a multi-dimensional trajectory, composed of an input trajectory and an output trajectory. It also fails to learn if the input trajectory is considered as prior knowledge and only the output Δm_z is the target trajectory without concatenating the extra delayed states of the input parameter and delayed output state of Δm_z (which corresponds to our results in the case $k = 1$).

For clarification, we have added this explanation in the main text,

'In that case, the test inputs are composed of a pulse signal of ΔK_u (or ΔD) and a constant value of D (or K_u) to induce an oscillating magnetic response Δm_z and thus to predict the corresponding frequency for specific material parameters D and K_u . They are thus different from the sinusoidal waveforms of the ΔD and ΔK_u used for training (Fig. 1d), which is important to test the ability of the neural network to generalize (see Methods)'

and in the Methods section,

'For the testing set of Mackey-Glass prediction task (results in Fig. 3), the input is a time varying ΔK_u in the form of preprocessed MG time series with a time interval of 10 ps (as shown in Suppl. Fig 5).'

'we build a single-trajectory training set \mathbf{y}_{true} consisting of n data points sampled from the output trajectory Δm_z for the skyrmion system and from ΔV_{out} for the experimental oscillator with a time interval Δt '

'The number of training data points is $n = 10,000, 15,000, 10,000,$ and $50,000$ and validation data points of $5,000, 5,000, 5,000,$ and $10,000$ are used for the one skyrmion system, the multi-skyrmions system, the parameter-based system and experimental data of oscillator, respectively.'

'The total number of testing points is $n = 800$ for each of the 37 different values of D and K_u in Fig. 1, $n = 500,000$ for the Mackey-Glass time series, $n = 522,8800$ for the experimental oscillator with the cochlear method, and $n = 850,7200$ for the experimental oscillator with the spectrogram method.'

Q6 (Clarification) Algorithm 1: it seems that the authors just train the neural ODEs with training data by using Algorithm 1 and then use the trained model to perform the prediction task. So there is no training algorithm designed specifically for prediction in future.

Yes, in our method, the prediction is achieved by setting an initial value and giving time-varying inputs into the trained Neural ODE. So, there is no algorithm specifically designed for prediction. We have clarified the prediction procedure method section in the main text, which now reads,

'The prediction is made by specifying an initial value of \mathbf{y} and applying the time-varying inputs $\mathbf{e}(t)$ into the trained Neural ODE. The test set is used for the evaluation of the prediction performance of the trained Neural ODE.'

Q6 Confusing terms. Neural ODE, NeurODE, SPIN-NEURAL ODE: are they indicating the same thing?

In the original manuscript, Neural ODE, NeurODE, SPIN-NEURAL ODE indeed all indicated the same thing. Spin-Neural ODE was used for referring to our modified version of Neural ODE for training input and output physical system, and NeurODE was used in the graphs as an abbreviation of 'Neural ODE'.

We have now clearly defined 'NeurODE' in the caption of the Figures, and, for clarity, unified the name of Spin-Neural ODE and Neural ODE as just Neural ODE.

Q7 In conclusion, I believe the authors should make it clear that the novelty of the manuscript is in that the state-of-the-art deep-learning technique for time-series modeling has been applied to complex real-world physical processes, but not designing a novel extension of NODEs.

We have significantly overhauled our manuscript, based on the suggestions of the reviewer and hope that the novelty of the work is now well clarified. In addition to the modifications presented until now, we have added the following sentence to make the contribution of the paper very explicit:

- In the introduction section.

'In the rest of the paper, we first explain how we modified Neural ODE in order to be able to train the whole set of parameters based on the temporal evolution of a single physical variable of the nanodevice under the effect of fluctuating inputs. For this purpose, we have integrated in the Neural ODE framework the idea of the embedding theorem for the reconstruction of the state space from a time series.'

'we demonstrate that this state-of-art deep-learning technique for time series modeling can be applied to a complex real-world physical processes. We train Neural ODEs to predict the results of real experiments on spin-torque nano-oscillators'

- In the discussion section.

'we have presented an efficient modeling approach for physical ODE-based systems, and highlighted its excellent performance on modeling real-world physical dynamics.'

'In particular, we demonstrate its use in modeling complex physical processes in the field of spintronics, which is considered one of the most promising future technologies for memory and computing.'

REVIEWERS' COMMENTS

Reviewer #1 (Remarks to the Author):

The authors addressed the comments of both reviewers, and the manuscript is recommended for publication.

Reviewer #2 (Remarks to the Author):

The authors satisfactorily answered all the comments and questions and suitably revised the manuscript. I believe the paper is appropriate for publication in Nature Communications.

Reviewer #3 (Remarks to the Author):

The authors addressed all the issues raised previously by the referee; as mentioned in the first round of review, the mathematical formulation of neural ODEs and their variants covered in the manuscript is sound and the contributions of this manuscript is now clearly shown.